# AZGP1 in POMC neurons modulates energy homeostasis and metabolism through leptin-mediated STAT3 phosphorylation

Sheng Qiu[1,2,7], Qinan Wu [3,7], Hao Wang[1,7], Dongfang Liu[1], Chen Chen [4], Zhiming Zhu [5], Hongting Zheng [6], Gangyi Yang [1]✉, Ling Li [1,2]✉ & Mengliu Yang [1]✉

Zinc-alpha2-glycoprotein (AZGP1) has been implicated in peripheral metabolism; however, its role in regulating energy metabolism in the brain, particularly in POMC neurons, remains unknown. Here, we show that AZGP1 in POMC neurons plays a crucial role in controlling whole-body metabolism. POMC neuron-specific overexpression of *Azgp1* under high-fat diet conditions reduces energy intake, raises energy expenditure, elevates peripheral tissue leptin and insulin sensitivity, alleviates liver steatosis, and promotes adipose tissue browning. Conversely, mice with inducible deletion of *Azgp1* in POMC neurons exhibit the opposite metabolic phenotypes, showing increased susceptibility to diet-induced obesity. Notably, an increase in AZGP1 signaling in the hypothalamus elevates STAT3 phosphorylation and increases POMC neuron excitability. Mechanistically, AZGP1 enhances leptin-JAK2-STAT3 signaling by interacting with acylglycerol kinase (AGK) to block its ubiquitination degradation. Collectively, these results suggest that AZGP1 plays a crucial role in regulating energy homeostasis and glucose/lipid metabolism by acting on hypothalamic POMC neurons.

Obesity and insulin resistance (IR) are important causes of metabolic diseases, such as the very common diseases type 2 diabetes (T2D) and nonalcoholic fatty liver disease[1–3]. The true cause of obesity is an imbalance toward energy intake and away from energy consumption, which is primarily controlled by the central nervous system (CNS), especially the hypothalamus. The regulatory effect of the hypothalamus on food intake, body weight (BW), and energy homeostasis involves a complex process, and the arcuate nucleus (ARC) of the hypothalamus is the center of the neural network responsible for energy regulation[4]. The ARC contains two opposing neuronal populations: orexigenic/agouti-related peptide (AgRP)/neuropeptide Y

(NPY)-containing neurons and anorexic proopiomelanocortin (POMC)-derived peptide-containing neurons[1,5]. These metabolic regulatory neurons can sense changes in nutrient, metabolite, hormone, and cytokine levels and integrate these signals to regulate body metabolism. POMC neurons suppress food intake and BW gain by antagonizing the orexigenic activities of AgRP/NPY neurons, in part by releasing α-melanocyte-stimulating hormone[6–8]. Additionally, these neurons regulate energy metabolism by changing sympathetic nervous system (SNS) activity or leptin sensitivity[9,10]. Recently, several neuropeptides/cytokines, such as Nesfatin-1 and mesencephalic astrocyte-derived neurotrophic factor, have been shown to affect

[1]Department of Endocrinology, the Second Affiliated Hospital, Chongqing Medical University, Chongqing 400010, China. [2]Key Laboratory of Medical Diagnostics of Ministry of Education, Department of Laboratory Medicine, Chongqing Medical University, Chongqing 400016, China. [3]Department of Endocrinology, The Affiliated Dazu Hospital of Chongqing Medical University, Chongqing 402360, China. [4]Endocrinology, SBMS, Faculty of Medicine, University of Queensland, Brisbane, QLD 4072, Australia. [5]Department of Hypertension and Endocrinology, Daping Hospital, Third Military Medical University, Chongqing 400042, China. [6]Department of Endocrinology, Xinqiao Hospital, Third Military Medical University, Chongqing 400037, China. [7]These authors contributed equally: Sheng Qiu, Qinan Wu, Hao Wang. ✉e-mail: gangyiyang@hospital.cqmu.edu.cn; liling@cqmu.edu.cn; mengliu.yang@cqmu.edu.cn

energy metabolism by regulating the activities of neuronal populations in different hypothalamic regions. Disrupting the function of these neuropeptides/cytokines in the hypothalamus leads to the dysregulation of energy metabolism[11-13]. Consequently, these neuropeptides/cytokines are regarded crucial for energy metabolism. Therefore, it is very important to study neuropeptide/cytokine pathways and the underlying molecular mechanism by which downstream signaling pathways are involved in metabolic regulation.

Zinc-alpha2-glycoprotein (AZGP1) is a 43-kDa protein that was initially isolated from humans[14]. The *Azgp1* gene is located on chromosome 7q22.1[15] and is expressed in many tissues and organs in rodents, such as liver, fat, lung, prostate, and adipose tissue[16,23]. In previous investigations, the circulating levels and mRNA and protein expression of AZGP1 in adipose tissue and the liver were significantly lower in T2D patients than in healthy controls[17,18]. In animal studies, AZGP1 expression was decreased in *ob/ob* mice and diet-induced obese (DIO) mice[19]. Furthermore, *Azgp1*-deficient mice easily gained weight, while transgenic mice overexpressing *Azgp1* had a decreased BW[20]. In vitro and in vivo studies revealed that AZGP1 regulated energy homeostasis[21,22], stimulated lipolysis and inhibited lipogenesis through the cAMP pathway in the liver and adipose tissue[23]; it also reduced blood glucose levels and increased insulin sensitivity[24,25]. A recent study found that AZGP1 mRNA and protein are present in brain tissues[17]. Therefore, AZGP1 may also be expressed and released by AZGP1-containing neurons. As a metabolism-related hormone, AZGP1 in the CNS, especially in the hypothalamus, may play a critical role in regulating whole-body metabolism and energy homeostasis. Unfortunately, the detailed physiological role of AZGP1 in hypothalamic metabolism-related neurons and the underlying mechanism remain unknown.

In this study, we demonstrated that AZGP1 expression was downregulated in the hypothalamus of obese mice. AZGP1 positively regulates leptin-JAK2-STAT3 signaling and increases leptin sensitivity in POMC neurons, thus promoting weight loss and energy expenditure. Mechanistically, we revealed that AZGP1 enhanced the leptin-JAK2-STAT3 signaling pathway by interacting with acylglycerol kinase (AGK) to block its ubiquitination degradation.

## Results

### AZGP1 is associated with obesity in humans and mice

To explore the relationship between AZGP1 and obesity in humans, circulating AZGP1 levels were measured in overweight (BMI 24–27.9 kg/m²), obese (BMI ≥ 28 kg/m²) and normal-weight (BMI 18.5–23.9 kg/m²) individuals. We found that the circulating AZGP1 levels in overweight/obese individuals were considerably lower than those in controls (Supplementary Fig. 1a). Correlation analysis revealed that AZGP1 was adversely correlated with BMI (Supplementary Fig. 1b). These findings indicate that AZGP1 may be physiologically involved in maintaining normal BW. Next, we examined the expression of AZGP1 protein in the hypothalamus of C57BL/6J (WT) mice of different ages and found that AZGP1 expression was lower at postnatal days 1 (P1) to 20 (P20) and in elderly mice (18 and 24 months), while higher expression was observed in mice aged 1–12 months. Therefore, these results suggest that AZGP1 expression in the hypothalamus is age dependent (Fig. 1a). The current study was performed in mice aged 8–12 weeks, a period coinciding with abundant AZGP1 expression in the hypothalamus.

To investigate the role of AZGP1 in energy homeostasis, AZGP1 mRNA and protein expression in lean compared with DIO or genetically obese mice was analyzed. Immunofluorescence (IF) staining showed that AZGP1 was most strongly expressed in metabolism-related hypothalamic nuclei, the ARC and the ventromedial hypothalamus (VMH). In addition, the number of AZGP1-positive neurons was significantly reduced in high-fat diet (HFD)-fed mice (Fig. 1b). This result was further supported by reduced AZGP1 mRNA and protein

expression levels in the hypothalamus and isolated mediobasal hypothalamus (MBH), including the ARC and VMH, of HFD-fed mice and *ob/ob* mice (Fig. 1c–e).

Next, C57BL/6J (WT) mice were subjected to a fasting-refeeding protocol. The mRNA expression of *Azgp1* was significantly decreased in the hypothalamus as the fasting period increased (from 12 h to 36 h) (Fig. 1f). AZGP1 expression was rapidly restored in the hypothalamus and isolated MBH samples of fasted mice after refeeding (Fig. 1f, g), suggesting that the AZGP1 level in the hypothalamus was closely related to nutrition status (feeding). Taken together, these results indicated that AZGP1 is differentially expressed in the hypothalamic MBH and is important for energy balance. Changes in energy status result in alterations in AZGP1 expression in the hypothalamus.

### Overexpression of AZGP1 in the hypothalamus ameliorates DIO

To determine whether AZGP1 is involved in the central control of energy homeostasis in vivo, adeno-associated viruses encoding *Azgp1* or GFP (AAV9-*Azgp1*/GFP) were injected into the MBH of 8-week-old male WT mice (Fig. 2a). The location of the injection site and efficacy of virus transduction were confirmed by IF staining (Fig. 2b). As expected, AZGP1 protein was highly expressed in the hypothalamus after injection (Fig. 2c).

To further evaluate the effects of hypothalamic AZGP1 on metabolic phenotypes, mice were fed a normal chow diet (NCD) or a HFD for 12 weeks (Fig. 2a). Upon HFD feeding, *Azgp1*-overexpressing mice had a reduced BW and decreased adiposity (Fig. 2d–g). This phenotype was associated with decreased energy intake and increased body temperature and energy expenditure (Fig. 2h–k). However, the BW, energy intake, rectal temperature and energy expenditure of the hypothalamic *Azgp1*-overexpressing mice fed an NCD diet were comparable to those of the control mice (Fig. 2d, e and h–j). Furthermore, in agreement with the increase in energy expenditure observed in HFD-fed *Azgp1*-overexpressing mice, there was a notable decrease in the number of lipid droplets and increase in UCP1 expression in brown adipose tissue (BAT) (Fig. 2l–n). Accordingly, H&E staining of epididymal white adipose tissue (eWAT) showed that *Azgp1* overexpression in the hypothalamus significantly reduced the adipocyte volume but not cell number, as determined by DNA content analysis (Fig. 2o–q). The decrease in adipocyte volume observed in HFD-fed mice with hypothalamic AZGP1 overexpression suggested enhanced lipolysis. Consistently, the levels of phosphorylated protein kinase A (PKA) and hormone-sensitive lipase (HSL), two key regulatory enzymes for lipolysis, were also significantly increased in the eWAT of these mice (Fig. 2r). Consistent with the observed resistance to obesity, a fasting-refeeding experiment showed that HFD-fed mice injected with AAV-*Azgp1* into the MBH had lower fasting and postprandial blood glucose levels (Supplementary Fig. 2a). The glucose tolerance test (GTT) and insulin tolerance test (ITT) revealed that under HFD feeding, glucose tolerance and insulin sensitivity were improved in mice overexpressing AZGP1 compared with control mice (Supplementary Fig. 2b, c). Consistent with the overall improvement in glucose homeostasis, hepatic lipid deposition in HFD-fed mice with hypothalamic AZGP1 overexpression was largely reduced (Supplementary Fig. 2d, e). However, AZGP1 overexpression in the hypothalamus did not lead to alterations in blood glucose levels, insulin sensitivity, glucose tolerance or histological changes in the liver under NCD feeding (Supplementary Fig. 2b–e). These results suggest that overexpression of AZGP1 in the hypothalamus improves glucose and lipid metabolism and prevents the development of obesity induced by HFD feeding.

### AZGP1 signaling in POMC neurons regulates energy metabolism

POMC and AgRP/NPY neurons play important roles in the regulation of food intake and energy homeostasis in the hypothalamic ARC[26]. To investigate the neuronal population in which AZGP1 functions, we assessed the distribution of AZGP1 in hypothalamic ARC neurons. We

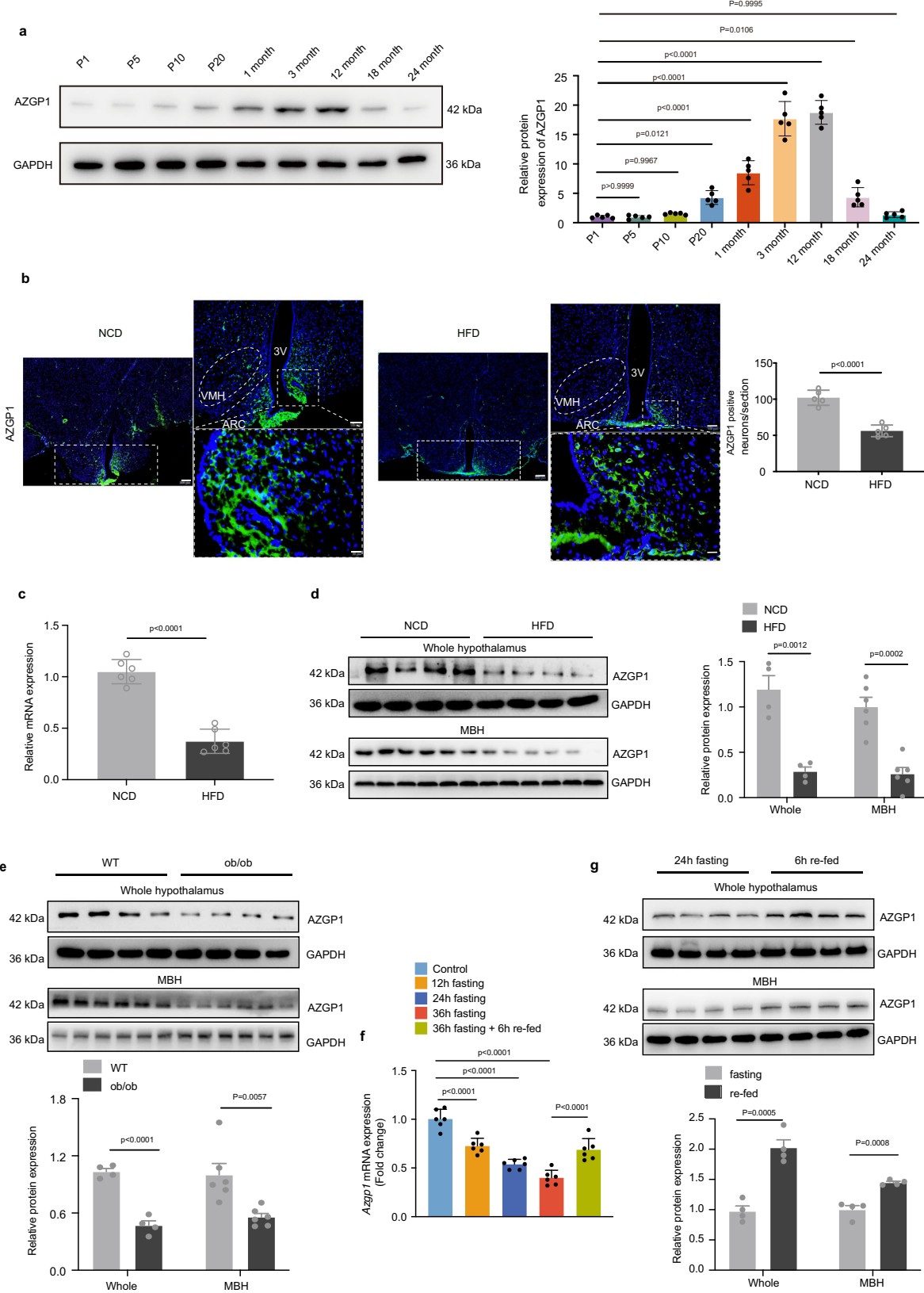

found that AZGP1 was widely expressed in these neurons, including those expressing POMC and AgRP (Fig. 3a). In the ARC, AZGP1 was expressed in 94.2% of POMC-expressing neurons and 86% of AgRP-expressing neurons (Fig. 3a). However, AZGP1 expression was rare in glial cells, as indicated by the labeling of AZGP1 with glial fibrillary acidic protein (GFAP) (Fig. 3a). Furthermore, the expression of AZGP1

in POMC neurons was lower in young and elderly mice but higher in adult mice (Supplementary Fig. 3).

To further evaluate the effects of AZGP1 on distinct neuronal populations, 8-week-old male WT mice were injected with AAV9-*Azgp1*/GFP into the MBH. Compared with that in MBH AVV9-GFP mice, AZGP1 overexpression resulted in a significant increase in *Pomc* mRNA

**Fig. 1 | AZGP1 expression is reduced in the hypothalamus of obese mice.**
**a** Immunoblotting analysis for AZGP1 protein expression in the hypothalamus of male mice pups (P1-20) and mice (1–24 months) (n = 5 mice). **b**–**d** Eight-week-old male WT mice were fed a NCD or HFD for 12 weeks. **b** Representative IF image of AZGP1 staining in the hypothalamus (n = 5 mice); scale bars: 200 μm (left), 100 μm (upper right), 20 μm (lower right). **c** *Azgp1* mRNA levels in the hypothalamus (n = 6 mice). **d** AZGP1 protein levels in the whole hypothalamus and mediobasal hypothalamus (MBH) (whole hypothalamus n = 4 mice; MBH n = 6 mice). **e** AZGP1 protein levels in the whole hypothalamus and MBH of WT and *ob/ob* mice (whole hypothalamus n = 4 mice; MBH n = 6 mice). *Azgp1* mRNA (**f**) and protein (**g**) expression in the whole hypothalamus and MBH of WT mice under fasting and refeeding conditions (n = 6 mice for mRNA; n = 4 mice for protein). 3V third cerebral ventricle, ARC arcuate nucleus, VMH ventromedial nucleus. The data are shown as the mean ± SEM. Two-tailed Student's *t* tests were used in (**b**–**e**) and (**g**), and one-way ANOVA followed by Tukey's test was used in (**a**) and (**f**). Source data are provided as a Source Data file.

expression in MBH AVV9-*Azgp1* mice, whereas there was no significant difference in *Npy* or *Agrp* mRNA expression (Fig. 3b). Immunohistochemical (IHC) staining showed that POMC expression in the ARC was significantly greater in MBH AAV9-*Azgp1* mice than in control mice, while AgRP expression remained unchanged (Fig. 3c). Next, mice were injected with leptin into the MBH to determine the effect of AZGP1 on the leptin-induced excitability of POMC neurons, as assessed by c-Fos staining. The results showed an increase in c-Fos staining intensity in POMC-expressing cells in MBH-AAV9-*Azgp1* mice compared to control mice (Fig. 3d). These findings imply that *Azgp1* overexpression in the hypothalamic ARC may increase POMC expression and neuronal activity.

To delineate the role of AZGP1 in POMC neurons, a Cre-dependent AAV9 expressing *Azgp1* or GFP (AAV9-DIO-*Azgp1*/GFP) was injected into the MBH of POMC-Cre mice to specifically overexpress AZGP1 in POMC neurons (POMC-*Azgp1*-OE mice), and the mice were fed an NCD or HFD for 12 weeks (Fig. 4a). GFP immunostaining was performed to confirm the injection site and efficacy of virus transduction in the ARC (Supplementary Fig. 4a). Overexpression of AZGP1 in POMC neurons reduced BW, fat mass, and energy intake, while increasing energy expenditure and body temperature in HFD-fed mice (Fig. 4b–i). Histological examination showed that AZGP1 overexpression in POMC neurons resulted in more BAT depots, fewer lipid droplets and increased *Ucp1* mRNA and protein expression in BAT under HFD feeding (Fig. 4j–m).

Previous studies have shown that activation of the SNS increases BAT thermogenesis[27–29]. Here, we found that the expression of tyrosine hydroxylase (*Th*), an enzyme involved in the synthesis of catecholamines, and a norepinephrine receptor (*Adrβ3*) was markedly higher in BAT from HFD-fed POMC-*Azgp1*-OE mice than in BAT from control mice fed the same diet, suggesting that activation of AZGP1 in POMC neurons increases SNS signaling (Fig. 4n). The volume of WAT was decreased, while there was no significant change in the number of adipocytes in adipose tissue (Fig. 4o–q). In addition, the phosphorylation of PKA and HSL in eWAT was significantly increased in HFD-fed POMC-*Azgp1*-OE mice compared with POMC-Cre mice fed the same diet (Fig. 4r), but there was no difference in the expression of fatty acid uptake- or lipogenesis-related genes in eWAT (Supplementary Fig. 4b). Furthermore, histological examination of the liver revealed a significant reduction in lipid deposition in POMC-*Azgp1*-OE mice compared with control mice fed a HFD (Supplementary Fig. 4c, d). These results indicate that *Azgp1* overexpression in POMC neurons promotes BAT thermogenesis and TG lipolysis in eWAT.

To assess the effect of AZGP1 signaling in the hypothalamus on glucose metabolism, a fasting-refeeding experimental protocol was performed. The results showed that fasting blood glucose levels (FBG, 0 h, and 12 h) and blood glucose levels 1–2 h after refeeding were significantly lower in HFD-fed POMC-*Azgp1*-OE mice than in control mice fed the same diet (Supplementary Fig. 4e), suggesting that glucose utilization was promoted by AZGP1 overexpression. Furthermore, HFD-fed POMC-*Azgp1*-OE mice had improved glucose tolerance and insulin sensitivity compared to their POMC-Cre littermates (Supplementary Fig. 4f, g), suggesting that POMC-AZGP1 signaling ameliorates glucose metabolism and insulin sensitivity. Notably, we found that when POMC-*Azgp1*-OE mice were fed a HFD for 4 weeks, a duration insufficient to alter their BW, the overexpression of AZGP1 did not

improve TG levels, glucose tolerance, or insulin sensitivity (Supplementary Fig. 4h–l). This finding suggested that the effect of AZGP1 on glucose/lipid metabolism in HFD-fed mice may be due to differences in BW. Furthermore, a similar metabolic phenotype was also observed in female mice (Supplementary Fig. 5a–n).

Next, we used a gain-of-function approach to investigate the role of AZGP1 in energy metabolism in AgRP neurons. To do this, AgRP-Cre mice were given bilateral injections of AAV9-DIO-*Azgp1*/GFP into the MBH to induce AZGP1 overexpression specifically in AgRP neurons (AgRP-*Azgp1*-OE) and fed either an NCD or HFD for 12 weeks (Supplementary Fig. 6a). The efficiency of virus transduction in the MBH was confirmed by GFP immunostaining (Supplementary Fig. 6b). Surprisingly, there were no differences in BW, energy intake, or energy expenditure between Agrp-*Azgp1*-OE mice fed a HFD or NCD and AgRP-Cre mice (Supplementary Fig. 6c–h). In addition, adiposity, liver steatosis, glucose tolerance, and insulin sensitivity were not altered by the overexpression of *Azgp1* in AgRP neurons (Supplementary Fig. 6i–o). Taken together, these results imply that *Azgp1* signaling in POMC neurons, but not in AgRP neurons, regulates whole-body energy metabolism.

### Inducible ablation of *Azgp1* in POMC neurons exacerbates DIO
Since the overexpression of AZGP1 in POMC neurons prevented the development of DIO, ablation of *Azgp1* in POMC neurons might exacerbate DIO. To avoid the developmental effects of *Azgp1* deletion, tamoxifen-induced POMC-Cre[ER] mice expressing tdTomato (POMC-Cre[ER]-tdTomato) were crossed with *Azgp1*[fl/fl] mice to generate mice with selective deletion of *Azgp1* in POMC neurons (POMC-*Azgp1* KO mice). In this model, Cre recombinase activity was controlled temporally to allow the deletion of *Azgp1* in adulthood[30] (Fig. 5a). IF staining for tdTomato (representing POMC neurons) and AZGP1 showed that AZGP1 was localized to POMC neurons in POMC-Cre[ER]-tdTomato mice but was almost completely absent in POMC-*Azgp1* KO mice (Supplementary Fig. 7a). As shown in Fig. 5b–d, BW and energy intake were significantly increased in POMC-*Azgp1* KO mice compared with those in *Azgp1*[fl/fl] mice (tamoxifen administered at 6 weeks of age) under HFD feeding, while energy expenditure and body temperature were dramatically reduced (Fig. 5e–g). Accordingly, these changes were accompanied by an increase in adipocyte hypertrophy and a decrease in PKA and HSL phosphorylation in eWAT (Fig. 5h–k). Furthermore, H&E staining revealed a significant increase in the size of adipocytes, along with a decrease in *Ucp1* mRNA and protein expression in the BAT of HFD-fed POMC-*Azgp1* KO mice (Fig. 5l–n). Similar to the alterations in adipose tissue, more lipid droplets were observed in the livers of POMC-*Azgp1* KO mice than in those of *Azgp1*[fl/fl] controls after 12 weeks of HFD feeding (Supplementary Fig. 7b, c). Moreover, the exacerbation of DIO in POMC-*Azgp1* KO mice was accompanied by compromised glucose tolerance and clearance (Supplementary Fig. 7d, e). Altogether, these results further demonstrated that AZGP1 signaling in POMC neurons regulate energy balance and glucose/lipid metabolism.

### The metabolic function of central AZGP1 is related to leptin signaling
Since leptin acts as the main contributor to appetite control by increasing POMC expression and inhibiting AgRP/NPY expression in the ARC[31], alterations in AZGP1 expression in the hypothalamus may

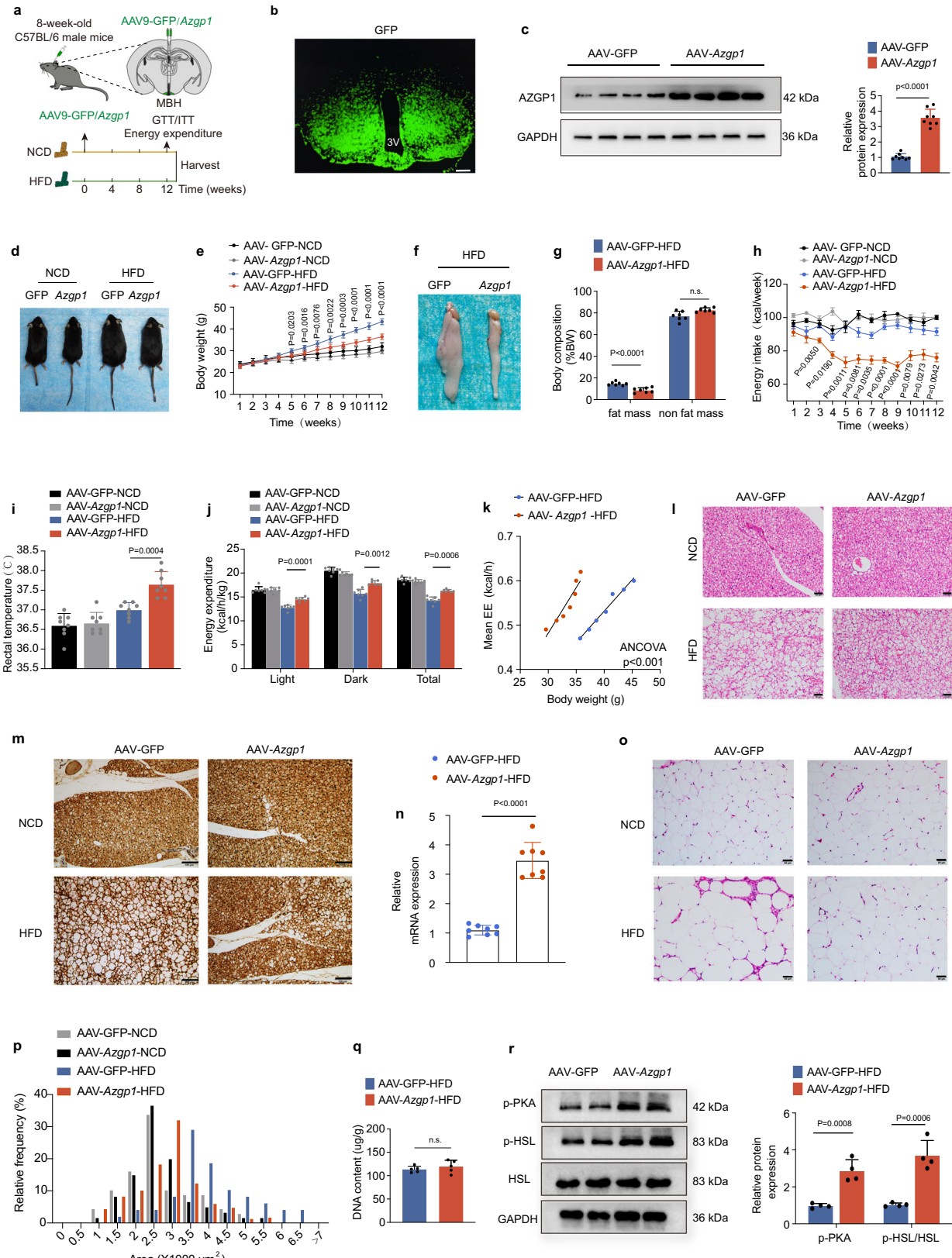

affect leptin receptor (ObRb) signaling. To confirm this, ObRb-Cre mice were given bilateral injections of AAV9-DIO-*Azgp1*/GFP into the MBH and fed a NCD or HFD for 12 weeks (Supplementary Fig. 8a). Postmortem IF staining for GFP confirmed that the ARC and VMH were effectively targeted by the virus (Supplementary Fig. 8b). Remarkably, in the DIO model, *Azpg1* overexpression within hypothalamic ObRb

neurons led to notable alleviation of obesity-like phenotypes and metabolic dysfunction (Supplementary Fig. 8c–p and 9a, b), including increased insulin sensitivity (Supplementary Fig. 9c, d). Conversely, *Azgp1* overexpression in hypothalamic ObRb neurons did not alter the metabolic phenotype in lean mice (Supplementary Fig. 8c, d, f, g, i, j, l–o). Moreover, IF staining showed that AZGP1 overexpression in ObRb

**Fig. 2 | Overexpression of AZGP1 in the mouse hypothalamus increases energy expenditure and diminishes the body weight gain in HFD-fed mice. a** Schematic representation of the experimental procedure. **b** Representative IF images showing GFP expression in the hypothalamus ($n = 5$ mice; scale bars: 200 μm). **c** AZGP1 protein expression in the hypothalamus ($n = 8$ mice). **d** A representative photograph of NCD- and HFD-fed mice ($n = 5$ mice). **e** Body weight curves of NCD- and HFD-fed mice ($n = 7$ mice). **f** Representative images of eWAT depots in HFD-fed mice ($n = 5$ mice). **g** Body composition in HFD-fed mice ($n = 7$ mice). **h** Energy intake ($n = 7$ mice). **i** Rectal temperature ($n = 8$ mice). **j** Energy expenditure ($n = 7$ mice). **k** Energy expenditure in HFD-fed mice ($n = 7$ mice). **l** Representative H&E staining image of BAT ($n = 5$ mice; scale bars: 50 μm). **m** UCP1 staining of BAT in NCD or HFD-fed mice ($n = 5$ mice); scale bars: 100 μm. **n** *Ucp1* mRNA expression in BAT ($n = 8$ mice). **o** Representative H&E staining image of eWAT ($n = 5$ mice; scale bars: 50 μm). **p** Cross- sectional areas of eWAT ($n = 5$ mice). **q** DNA content of eWAT in HFD-fed mice ($n = 5$ mice). **r** Western blot analysis of p-PKA and p-HSL/HSL levels in the eWAT of HFD-fed mice and densitometric analysis ($n = 4$ mice). 3V third cerebral ventricle, MBH mediobasal hypothalamus. The data are expressed as the mean ± SEM. Two-tailed unpaired Student's *t* test (**c**), (**g**), (**n**), (**q**, **r**) and one-way ANOVA followed by Tukey's test (**i**), or two-way ANOVA followed by Bonferroni's post hoc tests (**e**), (**h**), (**j**), one-way ANCOVA using body weight as covariate (**k**). Source data are provided as a Source Data file. (n.s. not significant.).

neurons increased MBH leptin-stimulated POMC expression and neuronal excitability in the ARC of HFD-fed mice, as indicated by an increase in the number of c-Fos-positive cells (Supplementary Fig. 9e, f). These results indicate that central AZGP1 is associated with leptin signaling and that AZGP1 overexpression in ObRb neurons elevates energy consumption by increasing the excitability of POMC neurons.

### AZGP1 regulates energy homeostasis *via* the leptin-JAK2/STAT3/POMC pathway

Leptin signaling increases POMC expression and suppresses AgRP expression *via* the Janus-activated kinase (JAK)-2 and signal transducer and activator of transcription (STAT3) pathways to regulate energy metabolism[31]. Thus, AZGP1 in POMC neurons may affect the leptin signaling cascade. To investigate this, HFD-fed POMC-*Azgp1*-OE or POMC-Cre mice were intraperitoneally injected with leptin (1 mg/kg) for 3 days. After leptin administration, BW and food intake decreased dramatically, especially in POMC-*Azgp1*-OE mice (Fig. 6a, b), suggesting that AZGP1 signaling in POMC neurons increased leptin sensitivity.

To further clarify the signaling pathway in which AZGP1 participates, leptin or artificial cerebrospinal fluid (aCSF) was infused into the MBH of HFD-fed POMC-*Azgp1*-OE or POMC-Cre mice (Fig. 6c). The effect of AZGP1 overexpression in the hypothalamus on the leptin signaling pathways was evaluated by western blotting (Fig. 6d). Consistent with the reductions in food intake and BW, the leptin-induced phosphorylation of JAK2 (pJAK2-Tyr1007/1008) and STAT3 (pSTAT3-Y705) was increased by AZGP1 overexpression in POMC neurons, while the phosphorylation levels of FoxO1 (Ser256) and mTOR (Ser2448) remained unchanged (Fig. 6d). Increased STAT3 phosphorylation was further confirmed by IHC staining (Fig. 6e). Similar results were observed in vitro. AZGP1 overexpression increased leptin-induced JAK2 and STAT3 phosphorylation in both the GT1-7 and Neuro2a (N2A) cell lines, whereas mTOR and FoxO1 phosphorylation levels remained unaffected (Supplementary Fig. 10a, b). Furthermore, leptin-induced POMC expression and the number of c-Fos-positive neurons were elevated by AZGP1 overexpression in HFD-fed mice, consistent with the increase in leptin sensitivity (Fig. 6f). In contrast, loss-of-function experiments revealed that ablation of *Azgp1* in POMC neurons attenuated leptin-induced STAT3 phosphorylation and POMC expression (Fig. 7a, b). These data demonstrated that AZGP1 in POMC neurons protects against DIO by promoting leptin-mediated JAK2/STAT3 phosphorylation and pSTAT3- induced POMC expression, as well as increasing the excitability of POMC neurons.

The electrical activity and excitability of POMC neurons are altered by the hunger-satiety cycle and energy balance; and some hormones often affect the excitability and electrical activity of these neurons to regulate energy balance[32]. Therefore, whole-cell current clamp recording of POMC neurons was performed to investigate the effect of specific ablation of *Azgp1* in POMC neurons on the electrophysiological properties of these neurons in HFD-fed mice. The baseline firing (spontaneous action potential, SAP) frequency of POMC neurons was significantly lower in POMC-*Azgp1* KO mice than in

Azgp1[fl/fl] mice (control), while the baseline resting membrane potential (RM) remained unchanged (Fig. 7c–e). Upon the administration of leptin to brain slices containing the MBH, an increase in the baseline firing frequency of POMC neurons was observed, and the deletion of *Azgp*1 in POMC neurons inhibited this leptin response (Fig. 7c, d). However, the RM was not obviously affected (Fig. 7e). Collectively, these results indicate that the regulation of metabolism by AZGP1 signaling may occur through a leptin-JAK2/STAT3-POMC-dependent mechanism in the hypothalamus.

### *Stat3* deletion in POMC neurons abolishes the effect of AZGP1 on metabolism

To obtain further insight into the role of STAT3 in the effects of AZGP1 on POMC neurons, *Stat3*[fl/fl] mice were crossed with POMC-Cre[ER]-tdTomato mice to generate tamoxifen-inducible POMC-*Stat3* KO mice, in which *Stat3* was deleted in POMC-expressing neurons after tamoxifen injection. Littermate POMC-Cre[ER] mice were employed as controls. The mice were intraperitoneally injected with tamoxifen at 6 weeks of age, then AAV9-DIO-*Azgp1*/GFP was injected into the MBH (Supplementary Fig. 11a). IF staining revealed that STAT3 (green) was localized in POMC neurons (tdTomato, red), whereas STAT3 localization in these neurons was significantly diminished in POMC-*Stat3* KO mice (Supplementary Fig. 11b, c). Under HFD feeding, the effects of AZGP1 overexpression in POMC neurons on BW, energy intake, energy metabolism, and glucose/lipid metabolism were completely abolished by *Stat3* deficiency (Supplementary Fig. 11d–u). These in vivo data further indicate that the protective effect of AZGP1 overexpression against DIO is mediated by the leptin-STAT3 signaling pathway in POMC neurons.

### AZGP1 regulates leptin-JAK2-STAT3 signaling by interacting with AGK

A previous study using immunoprecipitation and mass spectrometry (IP-MS) analysis suggested that AZGP1 interacts with AGK[33]. To investigate the role of AGK in DIO, we examined AGK expression in the hypothalamus of WT mice. IHC staining showed that AGK expression was significantly lower in the hypothalamic ARC of mice fed a HFD than in those fed a NCD (Supplementary Fig. 12a). To elucidate the effect of AZGP1 overexpression on AGK in the hypothalamus, AAV9-*Azgp1*/GFP was injected into the MBH of 8-week-old male WT mice, which were then fed a HFD for 12 weeks. Western blotting and IF staining demonstrated that AZGP1 overexpression significantly increased AGK protein expression in the hypothalamus, especially in the MBH (Fig. 8a, b). In contrast, deletion of *Azgp1* in POMC neurons significantly inhibited AGK expression in HFD-fed POMC-*Azgp1* KO mice (Supplementary Fig. 12b). Based on these results, we hypothesize that AGK plays a key role in the influence of hypothalamic AZGP1 on metabolic phenotype. To test this hypothesis, WT mice were injected bilaterally with AAV-*Azgp1*/GFP or AAV-*Azgp1* + AAV-*shAgk* into MBH and fed a HFD for 4 weeks. As shown in Supplementary Fig. 12c, d, *Agk* knockdown in the MBH under leptin stimulation completely eliminated the effects of central AZGP1 on food intake and BW. Furthermore, *Agk* knockdown also eliminated

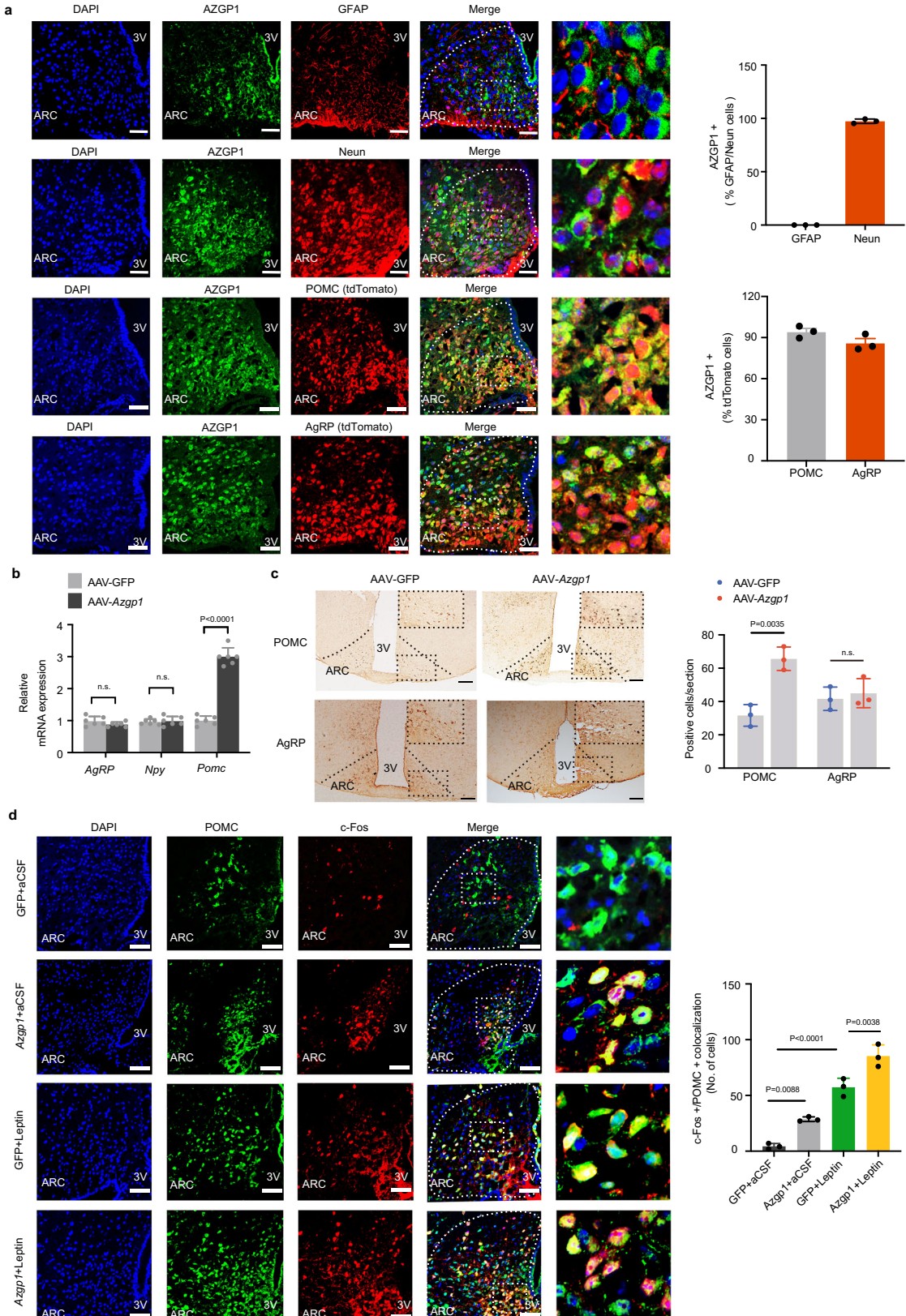

**Fig. 3 | Overexpression of AZGP1 in the mouse hypothalamus promotes POMC expression and the excitability of POMC neurons. a** Hypothalamic slices from NCD-fed male WT and POMC/Agrp-Cre:tdTomato mice were subjected to immunostaining with anti-AZGP1, anti-GFAP, and anti-neuronal protein (NeuN) antibodies. IF staining for AZGP1 expression (left) and the percentage of AZGP1-expressing neurons (right) ($n = 3$ mice; scale bar: 50 μm). **b–d** Eight-week-old male WT mice were given an injection of AAV9-*Azgp1*/GFP with or without leptin into the MBH. **b** The mRNA expression of *Agrp* and *Pomc* in the hypothalamus ($n = 6$ mice). **c** IHC staining of POMC and AgRP in the mouse ARC ($n = 3$ mice; scale bar: 100 μm). **d** IF staining of POMC and c-Fos in the mouse ARC ($n = 3$ mice; scale bar: 50 μm). 3V third cerebral ventricle, ARC arcuate nucleus. The data are expressed as the mean ± SEM, two-tailed Student's unpaired t-test for (**b**–**c**), and one-way ANOVA, followed by Tukey's multiple comparison test for (**d**). Source data are provided as a Source Data file. (n.s., not significant.).

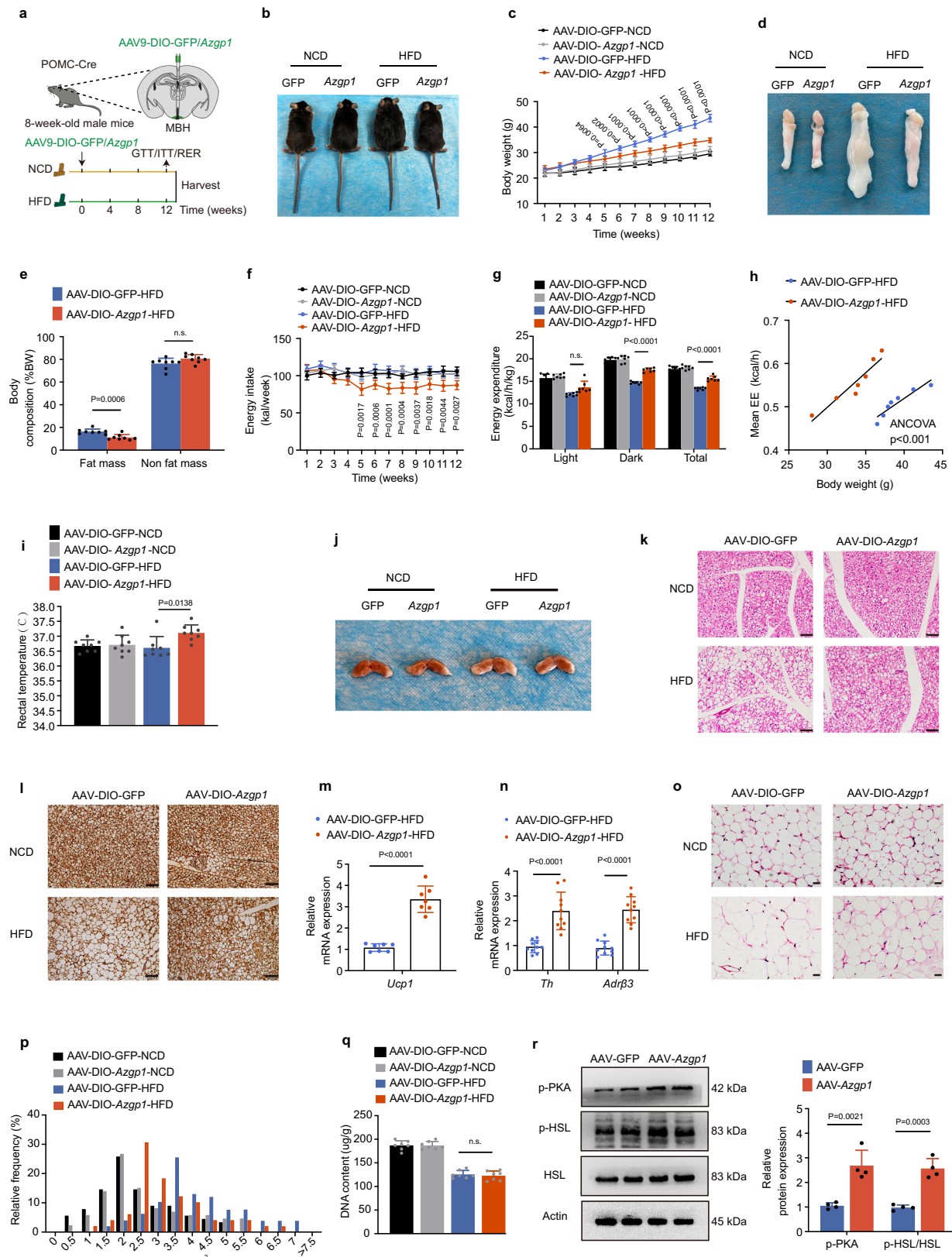

the effect of AZGP1 on JAK2 and STAT3 phosphorylation in the hypothalamus (Supplementary Fig. 12e). These effects were also observed in vitro (Fig. 8c). Interestingly, AZGP1 overexpression did not affect *Agk* mRNA expression in GT1-7 cells (Supplementary Fig. 12f), suggesting that AZGP1 is involved only in the post-transcriptional modification of AGK. To further characterize the

relationship between AZGP1 and AGK, the subcellular localization of AZGP1 and AGK was analyzed by IF staining. The AZGP1 signaling overlapped with the AGK signal in GT1-7 (Fig. 8d), N2A cells (Supplementary Fig. 12g), and the MBH in WT mice (Supplementary Fig. 12h), indicating that the two proteins were colocalized in these cells. However, whether AZGP1 interacts directly with AGK remains

**Fig. 4 | Specific overexpression of AZGP1 in POMC neurons increases energy expenditure and ameliorates metabolic disorders in HFD-fed male mice.**
**a** Schematic diagram of the experimental procedure. **b** Representative photograph of the mice ($n = 8$ mice). **c** Body weight curve ($n = 8$ mice). **d** Representative image of eWAT depots ($n = 3$ mice). **e** Body composition ($n = 8$ mice). **f** Energy intake ($n = 7$ mice). **g** Energy expenditure ($n = 7$ mice). **h** Energy expenditure ($n = 7$ mice). **i** Rectal temperature ($n = 8$ mice). **j** Representative image of BAT depots ($n = 5$ mice). **k** Representative H&E staining image of BAT ($n = 5$ mice; scale bar: 100 μm). **l** UCP1 immunostaining in BAT ($n = 5$ mice; scale bar: 100 μm). **m** *Ucp1* mRNA expression in BAT ($n = 7$ mice). **n** *Th* and *Adrß3* mRNA expression in BAT ($n = 10$ mice).

**o** Representative H&E staining image of eWAT ($n = 5$ mice; scale bar: 50 μm).
**p** Cross-sectional area of eWAT quantified by ImageJ analysis ($n = 5$ mice). **q** DNA content in total eWAT ($n = 7$ mice). **r** Western blot analysis of p-PKA and p-HSL/HSL expression in the eWAT of HFD-fed mice and densitometric quantification ($n = 4$ mice). MBH, mediobasal hypothalamus. The data are expressed as the mean ± SEM. Statistical significance was calculated using an unpaired two-tailed t-test (**e, m, n, q, r**) and one-way ANOVA followed by Tukey's test (**i**), or two-way ANOVA followed by Bonferroni's post hoc tests (**c, f, g**), one-way ANCOVA using body weight as covariate (**h**). Source data are provided as a Source Data file. (n.s. not significant.).

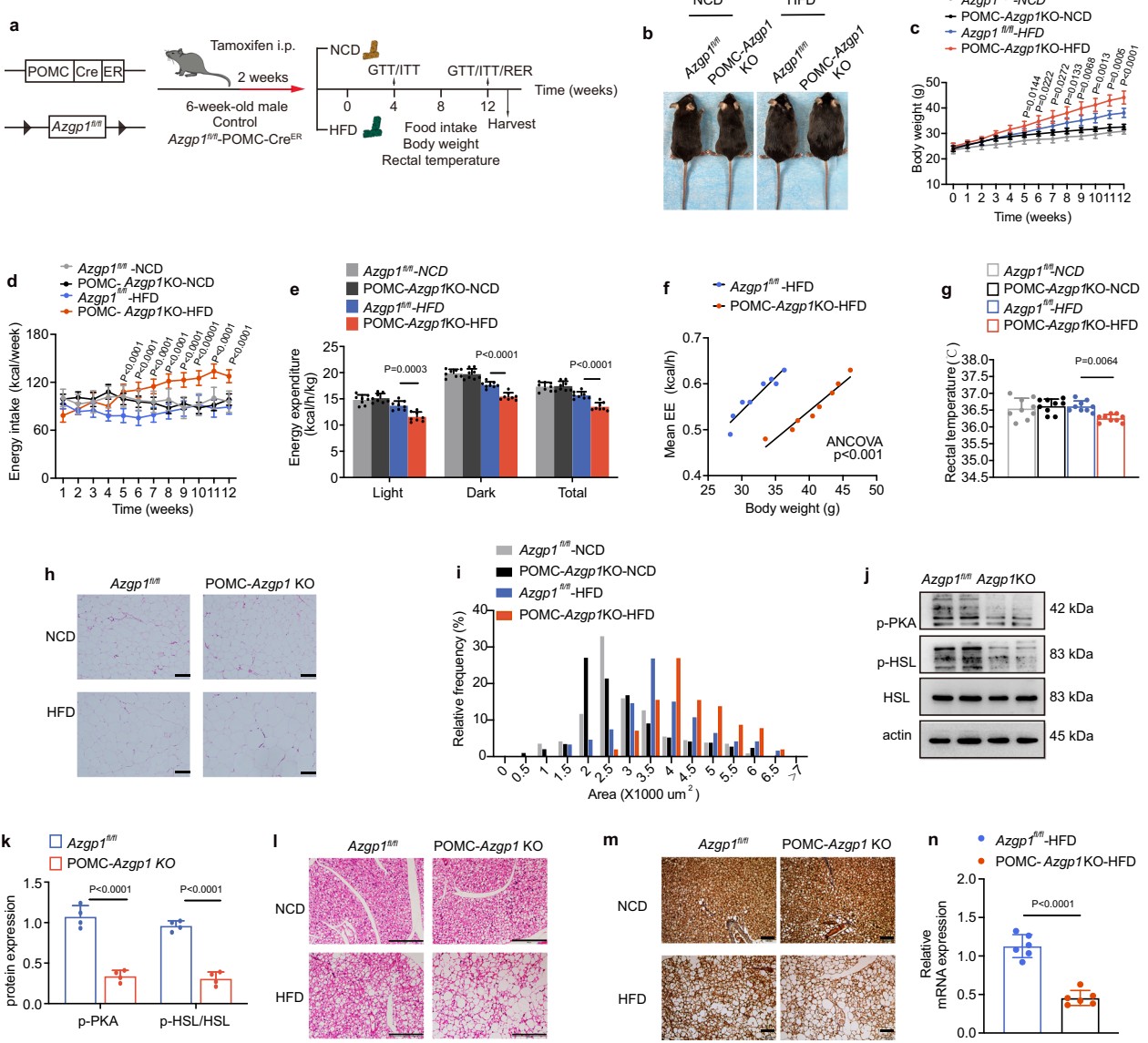

**Fig. 5 | Inducible ablation of *Azgp1* in POMC neurons increases susceptibility to DIO. a** Schematic diagram of the experimental procedure. **b** Representative image of 20-week-old mice ($n = 3$ mice). **c** Body weight curve ($n = 10$ mice). **d** Energy intake ($n = 8$ mice). **e** Energy expenditure ($n = 8$ mice). **f** Energy expenditure ($n = 8$ mice). **g** Rectal temperature ($n = 9$ mice). **h** H&E staining of eWAT ($n = 5$ mice); scale bars: 100 μm. **i** Cross-sectional area of eWAT quantified by ImageJ analysis ($n = 5$ mice). **j, k** Western blot analysis of p-PKA and t-HSL/p-HSL expression in eWAT of

HFD-fed mice and densitometric quantification ($n = 4$ mice). **l** H&E staining of BAT ($n = 5$ mice; scale bars: 200 μm). **m** UCP1 immunostaining of BAT ($n = 5$ mice; scale bars: 100 μm). **n** *Ucp1* mRNA expression in BAT ($n = 6$ mice). The data are expressed as the mean ± SEM. Statistical significance was calculated using a two-way ANOVA followed by Bonferroni's post hoc tests for (**c–e**); one-way ANCOVA using body weight as covariate (**f**); one-way ANOVA, followed by Tukey's test for (**g**); two-tailed Student's unpaired *t*-test for (**k, n**). Source data are provided as a Source Data file.

unknown. Subsequent co-IP experiments were performed in N2A and GT1-7 cells and confirmed that AZGP1 and AGK exist within the same protein complex and interact with each other (Fig. 8e–g).

Ubiquitination serves as a critical mechanism in the modulation of protein levels[34]. Considering the positive regulation of AGK by AZGP1, it was postulated that AZGP1 might modulate the stability of the AGK protein. The effect of AZGP1 on AGK protein stability was assessed by

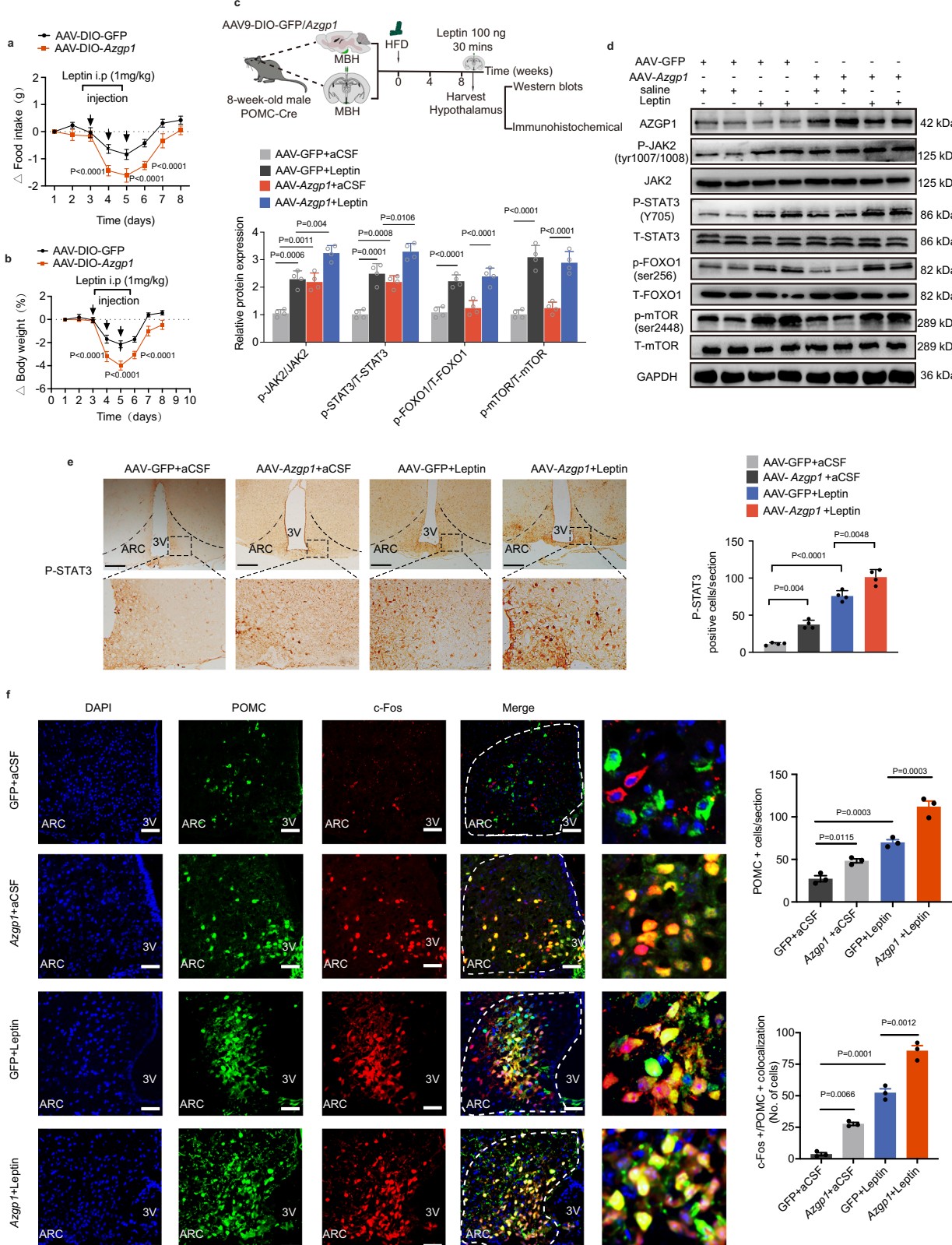

**Fig. 6 | AZGP1 regulates energy homeostasis via the leptin-JAK2/STAT3-POMC pathway. a, b** Eight-week-old male POMC-Cre mice were given an injection of AAV9-DIO-*Azgp1*/GFP into the MBH and then fed a HFD for 4 weeks. Leptin was then administered subcutaneously. **a** Changes in daily food intake ($n = 8$ mice). **b** Changes in body weight ($n = 8$ mice). **c** Schematic diagram of the experimental procedure. **d** Western blot analysis of total and phosphorylated JAK2, STAT3, FOXO1, and mTOR expression in the hypothalamus. GAPDH served as the loading control ($n = 4$ mice). **e** IHC staining of STAT3 phosphorylation in the ARC ($n = 4$ mice; scale bars: 100 µm). **f** IF staining of POMC and c-Fos in the ARC ($n = 3$ mice; scale bars: 50 µm). MBH mediobasal hypothalamus, 3V third cerebral ventricle, ARC arcuate nucleus. The data are expressed as the mean ± SEM. Statistical significance was calculated using an unpaired two-tailed *t*-test (**a, b**) and two-way ANOVA followed by Bonferroni's post hoc tests (**d–f**). Source data are provided as a Source Data file.

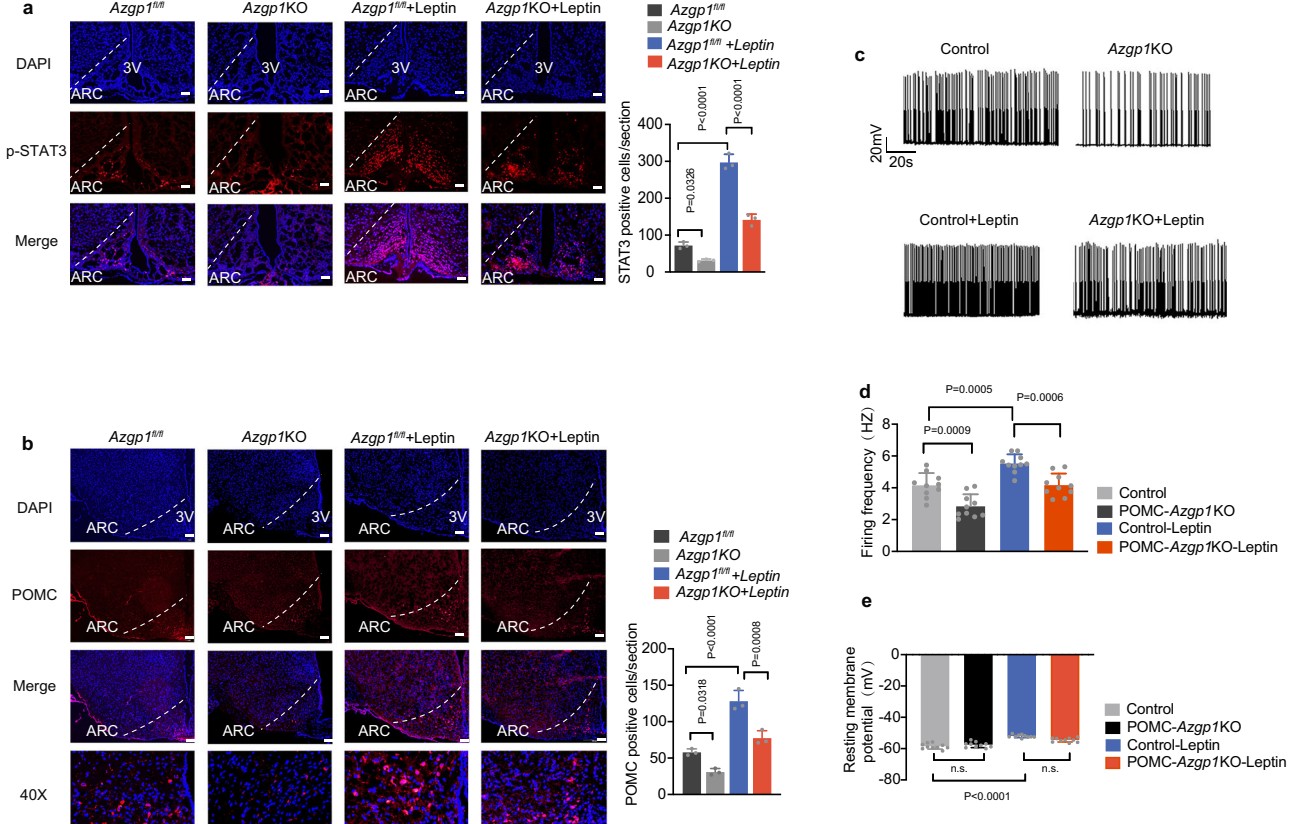

**Fig. 7 | Selective ablation of *Azgp1* in POMC neurons inhibits leptin-stimulated STAT3 phosphorylation and POMC neuron activity. a, b** Eight-week-old male POMC-*Azgp1* KO or *Azgp1*fl/fl mice were fed a HFD for 12 weeks and treated with or without leptin *via* MBH injection. IF staining of phosphorylated STAT3 (**a**) and POMC expression (**b**) in the ARC (n = 3 mice; scale bars: 200 µm). **c–e** POMC-*Azgp1* KO and POMC-Cre:tdTomato (control) mice were fed a HFD for 4 weeks. Representative current-clamp traces (**c**), average firing rate (**d**) and resting membrane potential (**e**) of POMC neurons (*n* = 10 mice). 3V third cerebral ventricle, ARC arcuate nucleus. The data are expressed as the mean ± SEM, two-way ANOVA for (**a–e**). Source data are provided as a Source Data file. (n.s. not significant.).

treating LV-*Azgp1*/GFP-transfected GT-1-7 and N2A cells with cycloheximide (CHX), a protein synthesis inhibitor. The results showed that the degradation speed of the AGK protein was reduced by *Azgp1* overexpression (Fig. 9a, b), while it was accelerated by *Azgp1* knockdown (Fig. 9c), indicating that AZGP1 affected the stability of the AGK protein.

Next, LV-*Azgp1*-transfected GT1-7 or N2A cells were treated with the proteasome inhibitor MG132 or the lysosome inhibitor 3-methyladenine A (3MA). The results showed that MG132 treatment significantly increased the expression of the AGK protein, while only a slight increase in AGK protein expression was observed in *Azgp1*-overexpressing cells (Fig. 9d, e). In contrast, *Azgp1* knockdown decreased the expression of AGK protein in N2A cells, while MG132 treatment blocked this effect (Fig. 9f). However, 3-MA treatment did not result in changes in AGK expression in N2A cells (Fig. 9g). Therefore, the data presented here reveal that AZGP1 modulates AGK stability in a proteasome-dependent manner.

To further determine the effect of AZGP1 on the ubiquitination of AGK, GT-1-7, and N2A cells were transfected with LV-*Azgp1*/GFP. The endogenous ubiquitination of AGK was evaluated by co-IP analysis. In *Azgp1*-overexpressing GT-1-7 and N2A cells, the ubiquitination of AGK was strongly suppressed (Fig. 9h, i). Similarly, exogenous ubiquitination analysis showed that AZGP1 inhibited the ubiquitination of AGK (Fig. 9j). Collectively, these findings demonstrate that during HFD feeding, AZGP1, which is downregulated, interacts with AGK to promote its proteasome-dependent ubiquitination and degradation, hence inactivating the downstream leptin-JAK2/STAT3-POMC pathway.

## Discussion

Leptin, a peptide hormone produced by adipocytes, is a key regulator of mammalian metabolism. Although several mechanisms have been proposed to explain leptin resistance, including gene mutations[35], defects in blood-brain barrier (BBB) permeability[36], inflammation[37] and endoplasmic reticulum (ER) stress[38], the relative contributions of these mechanisms to the impaired leptin response in the CNS remain unknown. In the current study, we demonstrated that AZGP1 in the hypothalamus plays a critical role in the control of food intake and energy homeostasis. AZGP1 regulated metabolism by modulating leptin- JAK2-STAT3 signaling *via* its interaction with the AGK protein. Moreover, AZGP1 suppresses the proteasome-dependent ubiquitination of AGK, which manipulates JAK2/STAT3 signaling in POMC neurons[39]. Therefore, the increase in AZGP1 expression in the hypothalamus increased neuronal leptin sensitivity, which resulted in resistance against DIO.

Although AZGP1 is an adipokine with multiple functions, its role in obesity and metabolic diseases is not fully understood[40]. Recently, AZGP1 was found to be expressed in brain tissue and plays a role in several CNS diseases, such as epilepsy and Alzheimer's disease[17]. However, the role of central AZGP1 in energy metabolism remains unknown. Here, we found that hypothalamic AZGP1 expression was reduced in obese mice, especially in the ARC, a key nucleus involved in maintaining normal energy homeostasis. AZGP1 expression in the hypothalamus was likewise dependent on nutrient status, as demonstrated by our fasting-refeeding experiment. The overexpression of AZGP1 in the MBH improved glucose/lipid metabolism and prevented the development of obesity. These evidences strongly support the

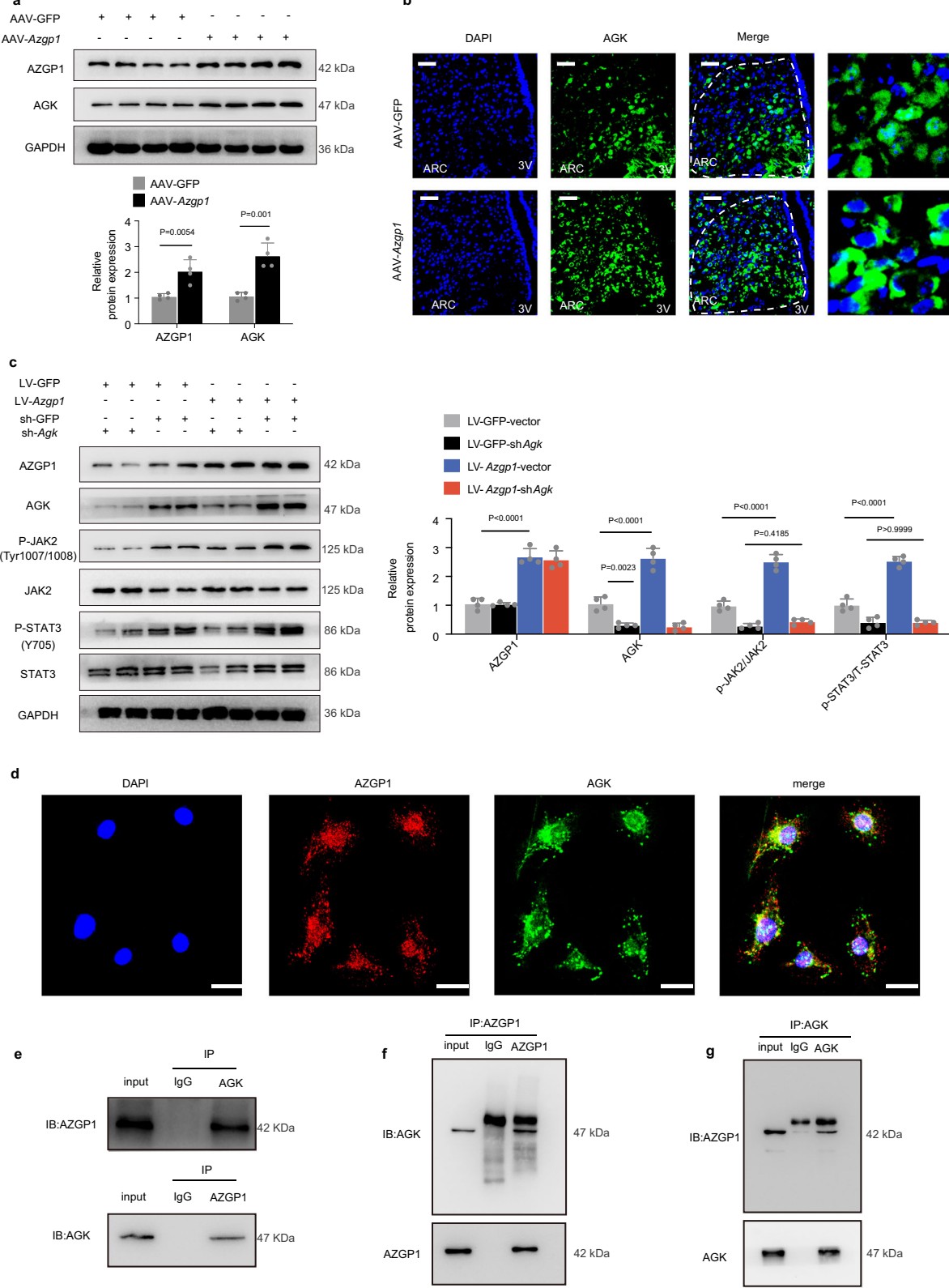

**Fig. 8 | AZGP1 regulates leptin-JAK2-STAT3 signaling by interacting with AGK.**
**a**, **b** Eight-week-old male WT mice were given an injection of AAV9-*Azgp1*/GFP into the MBH and fed a HFD for 12 weeks as described in the Methods. **a** AZGP1 and AGK protein expression in the hypothalamus (*n* = 4 mice). **b** IF staining of AGK in the hypothalamic ARC (*n* = 5 mice; scale bars: 50 μm). **c** GT1-7 cells were transfected with or without LV-*Azgp1*/GFP and/or LV-sh*Agk/vector* as described in the Methods. Western blots showing AZGP1, AGK, t-JAK2/p-JAK2 (Tyr1007/1008), and t-STAT3/p-STAT3 (Y705) expression (*n* = 4 independent cell experiments). **d** IF staining

showing the colocalization of AZGP1 and AGK in GT1-7 cells (*n* = 5 independent cell experiments; scale bars: 50 μm). **e–g** Cells were transfected with LV-*Agk* or LV-*Azgp1*. Cell extracts were subjected to immunoprecipitation (IP) with the specified antibodies. Co-IP analyses were performed in N2A (**e**) and GT1-7 (**f**, **g**) cells (*n* = 3 independent cell experiments). 3V third cerebral ventricle, ARC arcuate nucleus. The data are expressed as the mean ± SEM. Statistical significance was calculated using a two-tailed Student's unpaired t-test for (**a**), and two-way ANOVA followed by Bonferroni's post hoc tests for (**c**). Source data are provided as a Source Data file.

involvement of AZGP1 in the hypothalamic control of energy metabolism.

The hypothalamus is a dynamic brain region regarded as a master regulator of energy homeostasis. Neuronal populations in the hypothalamus are crucial for energy metabolism; in particular, the ARC contains two types of metabolism-related neurons: anorexigenic POMC and orexigenic AgRP/NPY neurons[6]. These neurons integrate various signals that convey the energy state of the periphery to maintain energy balance[41]. Therefore, identifying the type of neurons involved in the regulation of energy metabolism by hypothalamic AZGP1 signaling is important. Here, AZGP1 overexpression in hypothalamic POMC neurons was found to confer resistance to DIO, ameliorate glucose and lipid metabolism in the liver and promote lipolysis, browning of WAT, and energy expenditure in mice. Conversely, the deletion of *Azgp1* in POMC neurons increased susceptibility to obesity and aggravated metabolic dysfunction. Intriguingly, in AgRP neurons, AZGP1 signaling did not lead to a change in the metabolic phenotype under either NCD or HFD conditions. These results demonstrated that the POMC system is involved in the regulatory effect of central AZGP1 signaling on energy homeostasis and that central AZGP1 regulates metabolism in a POMC neuron-specific manner. Interestingly, a recent study challenged the prevailing view[29,42–46] by revealing that chronic activation of POMC neurons has minimal influence on preventing or reversing obesity[47]. We believe that this discrepancy may be due to the heterogeneity of POMC neurons and variations in activation methods, mouse models, and the diet/feeding period used in these studies[42,44,48,49]. Therefore, further investigation of how POMC neurons affect energy balance and the exact underlying mechanisms is needed.

Leptin plays an important role in the control of energy metabolism by modulating the excitability of POMC and AgRP neurons[50]. Here, we showed that the abundance of AZGP1 in the hypothalamus was altered by nutritional status. Fasting decreased AZGP1 expression, whereas feeding increased AZGP1 expression, which was consistent with the fluctuations in leptin levels[51]. Therefore, the effects of AZGP1 in the hypothalamus may be closely related to leptin signaling. In this study, ObRb-neuron-specific *Azgp1*-overexpressing mice were used. ObRb is widely expressed in the brain and plays a key role in leptin signaling as a receptor. ObRb-expressing neurons exist in several hypothalamic nuclei, including the ARC, VMH, lateral hypothalamic area (LHA), dorsomedial hypothalamus (DMH), and other nuclei that control energy homeostasis[52,53]. The ARC and VMH contain the primary neuronal populations associated with the metabolism-regulating actions of leptin and ObRb in vivo. Consistent with the effect of AZGP1 overexpression in POMC neurons, ObRb neuron-specific *Azgp1*-overexpressing mice exhibited a metabolically healthier phenotype in this experiment. These results indicate that the regulatory effect of AZGP1 signaling in CNS on energy homeostasis is closely related to leptin-receptor signaling in POMC neurons.

Recent evidence indicated that the loss of ObRb in POMC neurons did not affect energy balance in NCD-fed mice[54,55]. Consistent with these reports, the loss or overexpression of AZGP1 in POMC neurons had negligible effects on energy balance in the NCD-fed mice in this study. This may indicate that the physiological effects of interrupting AZGP1 signaling in animals with normal food consumption and weight are minimal and/or that AZGP1 signaling may be compensated for by other pathways.

Hypothalamic neurons respond to peripheral signaling, such as insulin and leptin signaling, to regulate energy balance. Physiologically, leptin binds with its receptor to activate JAK2, which leads to the phosphorylation and translocation of STAT3 and then promotes the excitability of POMC neurons. Therefore, leptin regulates energy metabolism through the JAK2/STAT3-POMC pathway[56]. Consistent with this notion, overexpression of AZGP1 in the hypothalamus promoted leptin-mediated phosphorylation of JAK2/STAT3 and increased POMC expression and neuronal excitability. In contrast, selective

ablation of *Azgp1* in POMC neurons decreased the firing frequency of POMC neurons. Based on these findings, we believe that central AZGP1 signaling promotes the effect of leptin on POMC neurons. However, the effect of AZGP1 on glucose/lipid metabolism may be a result of BW changes in these animals rather than the direct biological consequences of regulating AZGP1 expression in POMC neurons.

In a recent study, IP-MS analysis revealed that AZGP1 was associated with AGK in cultured neurons[33]. AGK is recognized as a common upstream molecule of the leptin-JAK2- STAT3 cascade and is capable of activating JAK2/STAT3 signaling[39]. Herein, we found that AGK knockdown blocked Azgp1-induced JAK2/STAT3 phosphorylation and that AGK and AZGP1 colocalized and interacted with each other. Mechanistically, AZGP1 binds to AGK and promotes its ubiquitination-mediated degradation. Therefore, the inhibition of AGK degradation by the AZGP1-AGK interaction is necessary for the regulatory action of AZGP1.

Nevertheless, it is worth noting that a previous study reported that AZGP1 interferes with anti-lipolysis function by binding to amine oxidase copper-containing 3 (AOC3)[57]. However, our in vitro and in vivo findings demonstrated that AZGP1 does not function through AOC3 in the hypothalamus (See Peer Review File). Therefore, we speculate that AZGP1 might operate through different mechanisms depending on the tissue or cellular context.

Circulating leptin levels are generally believed to be proportional to body fat mass. In obese animals, circulating leptin levels are significantly elevated[10]. Simultaneously, leptin signaling is inhibited in the hypothalamus due to reduced sensitivity of the leptin receptor, and exogenous leptin treatment may not reverse the obesity phenotype[10]. Therefore, the concept of leptin resistance has been proposed[2,6]. Recently, controversial opinions on the clinical benefit of increasing leptin sensitivity in the context of obesity have been given[58–60]. Several studies have indicated that augmenting the effects of leptin with certain chemicals may increase sensitivity to leptin and lead to weight loss[61]. Together, these observations and the current findings related to AZGP1 indicate that increasing leptin sensitivity could be a potential therapeutic strategy to combat obesity-associated metabolic disorders.

In summary, the current study demonstrated that HFD feeding led to the downregulation of AZGP1 expression in the hypothalamus, promoted the ubiquitination-mediated degradation of AGK and thus inactivated the downstream leptin-JAK2/STAT3-POMC signaling pathway, resulting in peripheral dysregulation of glucose/lipid metabolism and obesity-related phenotypes. This previously undiscovered role of AZGP1 in the hypothalamus indicates that it is a potential drug target for the treatment of obesity and its associated metabolic disorders (Fig. 9k).

## Methods
### Human samples
The work was approved by the Human Research Ethics Committee of Chongqing Medical University. Individuals were recruited from the Second Affiliated Hospital of Chongqing Medical University from 2018 to 2021 (project license number: XHEC-C-2012-023). None of the individuals took any medication or underwent a lifestyle intervention. Individuals with any other disease were excluded from the study. Individuals provided written informed consent. Serum AZGP1 levels were measured using an ELISA kit (Ray Biotech). The inter-assay and intra-assay coefficients of variation were <12% and <8%, respectively.

### Experimental animals
All procedures were approved by the Animal Experimentation Ethics Committee, Chongqing Medical University. Eight-week-old male C57BL/6 J (WT) mice were acquired from GemPharmatech Co., Ltd. (Jiangsu, China). POMC-Cre mice, tamoxifen-inducible POMC-Cre (POMC- Cre^ER) mice and Rosa26-tdTomato (tdTomato) reporter mice

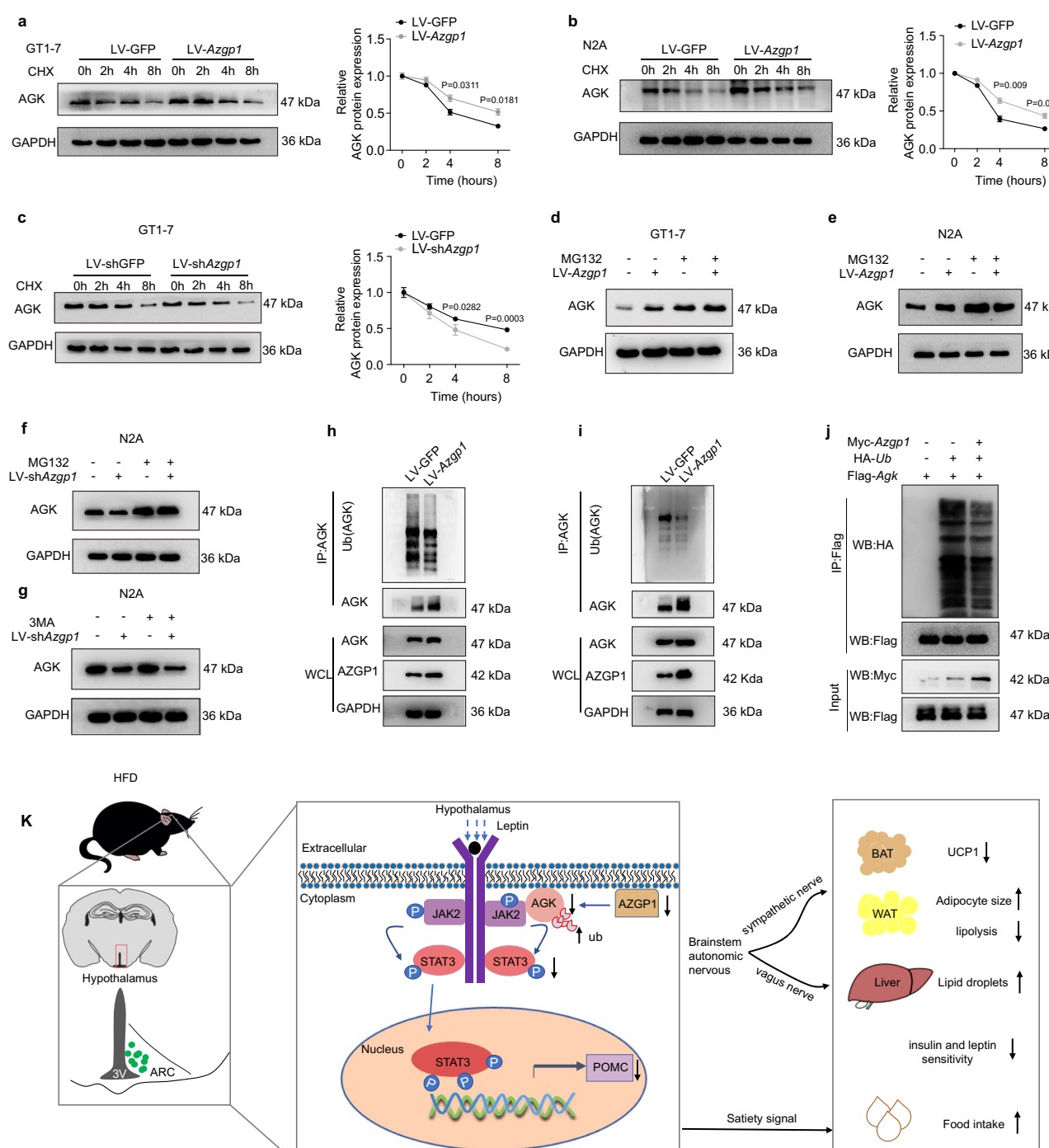

**Fig. 9 | AZGP1 inhibits the proteasome-dependent ubiquitination of AGK.**
**a**, **b** GT1-7 or N2A cells were transfected with LV-*Azgp1*/GFP and subsequently treated with or without cycloheximide (CHX, 20 μg/ml). The stability of the AGK protein was measured by western blotting in GT1-7 (**a**) and N2A (**b**) cells (*n* = 3 independent cell experiments). **c** GT1-7 cells were transfected with LV-*shAzgp1*/shGFP and treated with CHX (20 μg/ml) for the indicated times. The stability of the AGK protein was examined by Western blotting (*n* = 3 independent cell experiments). **d**, **e** GT1-7 or N2A cells were transfected with LV-*Azgp1*/GFP and treated with or without MG132 (40 μM). AGK protein expression was measured by Western blotting in GT1-7 (**d**) and N2A (**e**) cells (*n* = 5 independent cell experiments). **f**, **g** N2A cells were transfected with LV-*Azgp1*/GFP and treated with or without MG132 or 3-MA. AGK protein expression was measured by western blotting in MG132-treated

(**f**) and 3MA-treated cells (**g**) (*n* = 5 independent cell experiments). **h**, **i** GT1-7 and N2A cells were transfected with or without LV-*Azgp1*. Co-IP analyses were performed to evaluate the endogenous ubiquitination of AGK in GT1-7 (**h**) and N2A (**i**) cells (*n* = 3 independent cell experiments). **j** HEK293T cells were transfected with plasmids expressing Myc-*Azgp1* and/or HA-*Ub* and Flag-*Agk* as described in the Methods. Co-IP analyses were performed to assess the proteasome-dependent ubiquitination of AGK (*n* = 3 independent cell experiments). **k** Schematic illustration of cellular and molecular events underlying central AZGP1-mediated regulation of energy metabolism. P Phosphorylation, ARC arcuate nucleus, BAT brown adipose tissue, WAT white adipose tissue. The data are expressed as the mean ± SEM, two-tailed Student's unpaired *t*-test was used for (**a**–**c**). Source data are provided as a Source Data file.

were provided by Prof. Guo (Chinese Academy of Sciences, Shanghai, China). *Stat3*$^{flox/flox}$ mice were provided by Dr. Liu (Huazhong University of Science and Technology, Wuhan, China). *Azgp1*$^{fl/fl}$ (S-CKO-01376), AgRP-Cre (C001249), ObRb-Cre (C001036) and *ob/ob* mice were obtained from Cyagen Biotechnology Co., Ltd. (Jiangsu, China). All mice were on a C57BL/6 background. Housing conditions included a controlled temperature of 25 °C, humidity (40–60%), and a 12-h light/dark cycle, with water and food were provided. POMC-Cre$^{ER}$ mice were crossed with tdTomato reporter mice to generate mice with tdTomato-labeled POMC neurons (POMC-Cre$^{ER}$-tdTomato mice). *Azgp1*$^{fl/fl}$ mice or *Stat3*$^{fl/fl}$ mice were crossed with POMC-Cre$^{ER}$-tdTomato mice to produce POMC-*Azgp1* KO and POMC-*Stat3* KO mice expressing tdTomato, respectively.

To establish diet-induced obesity or IR animal models, 8-week-old male mice were fed a NCD (containing 15.75% of calories from fat, AIN-93G, Xietong Pharmaceutical Bio-engineering Co., Ltd;) or HFD (60% fat, D12492, Research Diets, New Brunswick, NJ) for 4 or 12 weeks. To induce adult-onset gene deletion in POMC neurons, 6-week-old male POMC-*Azgp1* KO or POMC-*Stat3* KO mice were injected with tamoxifen (0.15 g/kg) intraperitoneally (Sigma Aldrich, St. Louis, MO) for 14 days. Mice were euthanized in a carbon dioxide chamber at the designated time points during the study.

### Stereotaxic surgery and AAV injections
Mice were anesthetized with isoflurane and placed into a stereotaxic apparatus for stereotaxic surgery as previously described[62]. To induce *Azgp1* overexpression or *Agk* knockdown in the hypothalamus, WT mice were injected bilaterally with AAV9 expressing Azgp1/GFP (AAV-*Azgp1*/GFP) or *Azgp1*+sh*Agk* (AAV-*Azgp1* + AAV-sh*Agk*) (2 × 10$^{12}$ pfu/ml) into the MBH or ARC (−1.6 mm from bregma; ±0.3 mm from the midline; −5.80 mm from the dorsal surface) at a speed of 50 nl/min at 8 weeks of age. To induce the overexpression of *Azgp1* specifically in POMC, AgRP and ObRb neurons, POMC-*Stat3* KO, POMC-Cre, AgRP-Cre and ObRb-Cre mice were bilaterally injected with a Cre-dependent AAV expressing AZGP1 or GFP (AAV-DIO-*Azgp1*/GFP) (2 × 10$^{12}$ pfu/ml) into the ARC (−1.6 mm from bregma; ±0.3 mm from the midline; −5.80 mm from dorsal surface)[63]. The mice were fed either an NCD or an HFD for 12 weeks, starting 48 h after stereotaxic surgery. Food intake and BW were recorded daily or weekly. The location of the injection site and efficiency of virus transduction in hypothalamic nuclei were determined by IF staining of GFP. For the study of central leptin signaling, some mice were injected with leptin (100 ng, R&D Systems, Inc. Minneapolis) or aCSF into the MBH. Thirty minutes after leptin injection, hypothalamic tissues were collected following perfusion fixation.

### Metabolic phenotype analysis
Indirect calorimetry was performed by using an animal monitoring system (Clams; Oxymax, Columbus, USA), which measured various parameters, including BW, food intake, VO$_2$, VCO$_2$, and energy expenditure. A rectal probe connected to a digital thermometer was used to measure rectal temperature (Physitemp Instruments, NJ, USA).

### Leptin and insulin sensitivity analysis
For the leptin sensitivity test, 8-week-old male mice were fed a HFD for 4 weeks and given leptin (1 mg/kg) by intraperitoneal injection once daily for 3 days. Food intake and BW were monitored for 5 days. For the GTT and ITT, after 8 h of fasting, the mice were injected with glucose (2 g/kg) or insulin (1 U/kg) intraperitoneally. Blood samples were obtained from the tail tip at the indicated time points (0, 15, 30, 60, and 120 min).

### Histological and IHC analysis
Hematoxylin-eosin (H&E) and Oil Red O were used to stain WAT, BAT and/or liver tissues after fixation with 4% paraformaldehyde overnight[64]. For IHC staining, fat and brain tissue sections were dewaxed, rehydrated, and incubated in citrate buffer at 100 °C for 10 min. The tissue sections were incubated with primary antibodies overnight at 4 °C after being blocked in PBS with 5% goat serum. As previously reported[64], the tissue sections were then incubated with secondary antibody after washing with PBS. The antibodies that were used are listed in Supplementary Table S2.

### Hepatic triglyceride assay
The triglyceride concentration in the liver was measured using a Triglyceride Quantification Kit (A110-1-1, Jian Cheng Bioengineering Ins, Nanjing, China) according to the manufacturer's protocols.

### Cell culture and transfections
N2A cells (SCSP-5035) and HEK293T cells (GNHu44) were purchased from the National Collection of Authenticated Cell Cultures. GT1-7 cells (9Q01F4) were purchased from MERCK. N2A, GT1-7, and HEK293T cells were maintained in DMEM (Gibco) supplemented with 10% FBS and penicillin/streptomycin (100 ng/ml).

To overexpress Azgp1 in cells, lentiviruses expressing Azgp1 or GFP (LV-*Azgp1*/GFP) were transfected into N2A or GT1-7 cells for 48 h. To induce AGK knockdown in cells, LV-sh*Agk* was transfected into GT1-7 cells for 48 h. Leptin (200 ng/ml, R&D Systems, Inc., Minneapolis) or saline was administered to GT1-7 or N2A cells to study cell signaling. LV-*Azgp1*- or LV-*Agk*-transfected N2A and GT1-7 cells were used for the co-IP experiment. For the in vitro ubiquitination assay, N2A, GT1-7 or HEK293T cells were transfected with LV-*Azgp1*/ sh*Azgp1* or plasmids expressing Myc-*Azgp1* and/or HA-*Ub* and Flag-*Agk* for 48 h and treated with cycloheximide (CHX, 20 µg/mL, Aladdin) for 0, 2, 4, and 8 h, or MG132 (40 µM, Selleck, Houston, USA) for 12 h or 3MA (2 mM, MCE, New Jersey, USA) for 24 h.

### Hypothalamic MBH dissection
The MBH was microdissected as described in previous studies[65]. Briefly, the hypothalamic region was dissected from fresh sagittal sections of the brain (1 mm). The hypothalamic MBH was cut using an ultrathin blade following the mouse brain atlas. MBH was rapidly frozen in liquid nitrogen for further protein analysis[65].

### Immunofluorescence (IF) staining
For IF staining of tissues, brain sections were incubated with primary antibodies overnight at 4 °C, followed by incubation with secondary antibodies for 1 h at room temperature. For IF staining in cells, the cells were fixed with 4% formaldehyde for 20 min and permeabilized with 0.3% Triton X-100 for 20 min. Finally, the cells were cultured in blocking buffer and then stained as described above. A BX53F fluorescence microscope (Olympus, Center Valley, PA) or Nikon confocal microscope (Ti2E, Japan) was used to capture images of IF staining. The indicated antibodies are listed in Table S2.

### Western blotting
Protein samples extracted from N2A, GT1-7 or HEK293T cell lysates or adipose or hypothalamic tissues were separated by SDS–PAGE and transferred to PVDF membranes. The PVDF membranes were then immunoblotted with primary antibodies, as described previously[66]. The indicated antibodies are listed in Table S3. Densitometric analysis of the Western blot bands was performed using ImageJ software.

### Quantitative RT–PCR (qRT–PCR)
TRIzol reagent (Takara Biomedical Technology Co., Ltd, Beijing) was used to extract total RNA from adipose or hypothalamic tissue. Reverse transcriptase was used to synthesize complementary DNA. After combining cDNA, primers, and master mix, qRT–PCR was performed with the CFX Connect™ Optics Module (Bio-Rad, California, USA). The primer pairs used are shown in Table S4.

## Coimmunoprecipitation (Co-IP)

Co-IP analysis was performed as described previously[67]. In brief, N2A and CT1-7 cells were transfected with LV-*Azgp1* or LV-*Agk* for 48 h and then lysed with immunoprecipitation (IP) lysis buffer (Beyotime Biotechnology). The protein lysates were collected and centrifuged at 12,000 rpm for 15 min at 4 °C. The supernatants (500 μg protein) were immunoblotted with 5 μg of antibodies against AZGP1, AGK, or IgG (negative control) at 4 °C overnight. The protein antibody complexes were then incubated for 2 h at room temperature with A/G magnetic beads before being eluted with PBS containing 0.02% Tween 20. The immunoprecipitate was then centrifuged, and then immunoblotting was subsequently performed with the antibodies listed in Table S3.

## In vitro protein degradation and ubiquitination assays

The ubiquitination assay was performed as reported by ref. 68. The ubiquitination of AGK was examined in GT1-7, N2A, and HEK293T cells. The plasmids and lentiviruses described above were transfected into the cells for 48 h. Cell lysates were then collected and subjected to IP with an anti-AGK or anti-Flag antibody. The protein complexes were then subjected to Western blotting using an anti-Ub, anti-Flag, anti-Myc or anti-HA antibodies, as detailed in Table S3.

## Electrophysiological analysis

Six-week-old male POMC-Cre KO mice were intraperitoneally injected with tamoxifen for 14 days (0.15 g/kg), and the mice were fed a HFD for 4 weeks. Isoflurane was used to anesthetize the mice. The mice were then decapitated, and their brains were excised. The whole brains were rapidly sectioned (thickness of 300 μm) in ice-cold aCSF using a vibratome (Leica VT1200S) as described by ref. 69. The sections were recovered in a modified aCSF with or without leptin (300 nM) at 32 °C for 45 min before recording. Whole-cell patch clamp recording of tdTomato-labeled POMC neurons in the ARC was performed. These neurons were identified using an upright microscope equipped with an IR-DIC optical system (Eclipse FN-1, Nikon). SAP measurements were performed following the procedure reported by ref. 70. Neuronal SAPs at the RM were recorded in current-clamp mode by the whole-cell patch clamp technique. Electrophysiological signals were recorded and analyzed by a MultiClamp 700B amplifier (Axon Instruments) and Clampfit 10.3 software (Molecular Devices), respectively. These signals were filtered and digitized at 2 kHz and 10 kHz, respectively.

## Statistical analysis

SPSS software (version 25.0) or Prism software version 9.0 (GraphPad) was used for statistical analysis, and the data are presented as the mean ± SEM or mean ± SD. Differences among multiple groups were analyzed by one- or two-way analysis of variance (ANOVA) followed by Bonferroni's post hoc tests analysis or a two-sided unpaired Student's *t*-test, as appropriate. The comparison of EE between the groups was analyzed using one-way ANCOVA. Correlations between serum AZGP1 levels and BMI were performed using Pearson correlation analysis. $p < 0.05$ were considered statistically significant.

## Reporting summary

Further information on research design is available in the Nature Portfolio Reporting Summary linked to this article.

## Data availability

All data generated in this study are available within the article, Supplementary Information, or Source Data file. Source data are provided with this paper. Mouse brain atlas are available: https://searchworks.stanford.edu/view/9860513. Source data are provided with this paper.

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

## Acknowledgements
We thank Dr. Feifan Guo (Shanghai Institute of Nutrition and Health, Chinese Academy of Sciences, Shanghai, China) and Dr. Chaohong Liu (Huazhong University of Science and Technology, Wuhan, China) for kindly providing us with the POMC–Cre$^{ER}$ mice, AI9 (tdTomato) reporter mice and *Stat3*$^{fl/fl}$ mice. This work was supported by grants from the National Natural Science Foundation (82170816 and U22A20289 to L.L., 82300922 to S.Q., 82270853 to G.Y., and 82370852 to M.Y.), China Postdoctoral Science Foundation (2022MD713711 to S.Q.), Chongqing Natural Science Foundation Project—Postdoctoral Science Foundation Project (CSTB2022NSCQ-BHX0670 to S.Q.) and the CQMU Program for Youth Innovation in Future Medicine (W0159) to S.Q. and M.Y.

## Author contributions
S.Q., Q.W. and H.W. contributed equally. S.Q. conducted the animal experiments. Q.W. performed mechanistic studies. H.W. analyzed data. D.L., C.C. and Z.Z. contributed to technical assistance and discussion. H.Z. provided research material and technical assistance. G.Y. and L.L. wrote, reviewed, and edited the manuscript. M.Y. designed directed the project, and contributed to the discussion. G.Y., L.L. and M.Y. are the guarantors of this work and, as such, had full access to all the data in the study and take responsibility for the integrity of the data and the accuracy of the data analysis.

## Competing interests
The authors declare no competing interests.
