## [Peer Review File · Nature Communications]

AZGP1 in POMC neurons modulates energy homeostasis and metabolism through leptin-mediated STAT3 phosphorylationREVIEWER COMMENTS

Reviewer #1 (Remarks to the Author):

The manuscript by Qiu et al: Azgp1 in POMC neuron modulates energy homeostasis and 2 metabolism through leptin-mediated STAT3 phosphorylation is a great effort to link the glycoprotein Azgp1 to the leptin-mediated central control of metabolism and food intake. The major flaw of the manuscript is the rationale to consider the hypothesis of the ARC nucleus as a target of the plasma protein.

Even though Azgp1 has been found in CSF and brain in some neurological diseases, such as Krabbe or epilepsy, negative results have been found in control brain. Therefore, even though Azgp1 may improve leptin signaling acting as an antiinflammatory agent, since it may inhibit amine oxidase copper-containing 3 (Open Biol. 10: 190035.

<http://dx.doi.org/10.1098/rsob.190035>), the physiological expression of Azgp1 in the hypothalamus does not seem to be relevant. In fact the authors have employed stereotaxic surgery to allow the local expression of the protein.

In conclusion, even though the work could have been carried out correctly, unfortunately, there is not enough data in the literature to warrant this approach.

Reviewer #2 (Remarks to the Author):

Comments:

1. Line 31: reduced food intake and raised energy expenditure.
2. Line 38: I think you mean block ubiquitination and degradation.
3. Last paragraph needs a little more detail about the main findings and conclusion of the study.
4. Line 95: How was the blood assay validated. Levels are very high (mg/L range).
5. Fig 1C: Not clear what the 2 lower images are supposed to show. The bar graph panel is

not cited in the legend.

6. In the refeeding experiments, one might expect the opposite results, namely fasting raising Azgp1 and refeeding lowering based upon the findings with HFD and regular diet.

7. Many of the supplemental figures are primary, not supporting data and should be in the main manuscript (eg. Supp Fig 1 and 2).

8. Line 123-124: Should mention on normal calorie diet.

9. Line 125: Fig 1 panels cited do not match the text, which just talks about NCD.

10. Panel 1H: Why are animals on HFD eating less. Should express as energy intake rather than food weight.

11. Line 127: Body temperature results not mentioned.

12. For images showing UCP1 staining in BAT throughout the paper the differences in expression cited are not convincing. Would do RNA levels.

13. Line 136: PKA and HSL phosphorylation was affected, not total protein levels in 1Q.

14. While food intake is shown as total intake, energy expenditure is normalized to body weight, which may be misleading due to differences in BW between groups. Should show as linear regression plots of EE vs. BW. Temperature data however does tend to support what the authors state concerning EE.

15. Line 177: I do not see any WAT volume data shown.

16. Line 187: should read: effect of Azgp1 signaling in hypothalamus on glucose metabolism.

17. Line 239: Fig 6E does not show any NCD data.

18. Line 244: Hard to conclude leptin signaling per se affected as the baselines were different and increases after leptin were similar. Similar in Fig 6A, where leptin response was not affected, just baseline.

19. Line 282: I guess leptin was not administered to MBH but rather added to a brain slice including MBH.

20. Line 328: Why does AGK band pulled down by anti-Azgp1 run differently in panel 7E compared to input and IgG?

21. In Fig 8C the differences in degradation between the 2 conditions is not obvious. Should be quantified.

22. Line 363: In many assays leptin still shows responses and there are no direct data linking changes in AGK to effects on leptin signaling. Perhaps one could KO AGK and see if this blocks Azgp1 effects on leptin.

23. Line 420: Effects on insulin signaling were not addressed by experiments.

24. Line 427: Speculative, these changes could just be secondary to changes in adiposity.

Reviewer #3 (Remarks to the Author):

In this study, the authors have conducted extensive research to investigate the role of ZAG1 in POMC neurons using various biological and genetic approaches. The physiological consequences of ZAG1 over-expression or knockout were thoroughly examined, including food intake, energy expenditure, adipocyte size, glucose and insulin sensitivity. The authors conclude that Azgp1 regulates glucose/lipid metabolism through its action on hypothalamic POMC neurons. However, the effects of Azgp1 on glucose/lipid metabolism could be results of being obese or lean in these animals rather than the direct biological consequences of modulating ZAG1 expression in POMC neurons.

Interestingly, a recent publication titled "The melanocortin action is biased toward protection from weight loss in mice" suggested that over-expression of MC4R, POMC, or its derived peptides, which enhance melanocortin action, had little effect on preventing or reversing obesity. In contrast, the current study proposes that over-expression of ZAG1, which promotes POMC expression, ameliorates metabolic disorders in obese animals. This raises the question of the discrepancies between these findings.

To improve the manuscript, I recommend that the authors re-write and re-organize the data, addressing the following issues:

- 1) The novelty of the study is not clear, and there is redundant information in the main figure. For example, Figure 1 presents data on reduced circulating ZAG1 in human obesity and obese mice, which is already well-established. Consider moving this information to the supplementary figure if necessary.
- 2) Based on reference [17], ZAG1 expression appears to be age-dependent and cell type-dependent. What is the overall expression pattern of ZAG1 during development, particularly in POMC neurons?
- 3) In Figure 2, what percentage of POMC cells express ZAG1? Additionally, what percentage of AgRP cells express ZAG1?
- 4) In Supplementary Figure 8, it would be valuable to show the colocalization of ZAG1 with POMC and ZAG1 with cfos.
- 5) I suggest the authors re-write the discussion, specifically from line 422 to line 430, as the current study does not support the descriptions made. Again, the changes in glucose and lipid metabolism could be secondary effects of obesity.
- 6) The authors propose that the Azgp1-AGK interaction is necessary for the regulatory action of Azgp1. Is this regulatory mechanism specific to POMC neurons?

7) Rectal temperature was consistently measured in all experiments. What is the significance of the temperature changes in these animals? Does over-expression of *Azgp1* in POMC neurons modulate sympathetic output to thermogenic fat?

Other major issues.

Most of the IF figures are at very low resolution, it is very hard to ascertain the co-localization.

The animal numbers are limited in this study from 3 to 6, which is not enough for in vivo experiments.

What is the sex of the animals used in the experiments? Do the authors consider sex difference?

What's the circulating ZAG1 levels in the POMC-*Azgp1*-OE DIO mice.

In addition to these major issues, there are some minor concerns:

- Most of the immunofluorescence figures lack scale bars. Please add scale bars for better interpretation.
- Follow the HGNC guidelines for naming and formatting genes and proteins between humans and mice. Numerous mistakes in gene and protein names are present in the manuscript.

REVIEWER COMMENTS

Reviewer #1 (Remarks to the Author):

The manuscript by Qiu et al: Azgp1 in POMC neuron modulates energy homeostasis and metabolism through leptin-mediated STAT3 phosphorylation is a great effort to link the glycoprotein Azgp1 to the leptin-mediated central control of metabolism and food intake.

The major flaw of the manuscript is the rationale to consider the hypothesis of the ARC nucleus as a target of the plasma protein.

Even though Azgp1 has been found in CSF and brain in some neurological diseases, such as Krabbe or epilepsy, negative results have been found in control brain. Therefore, even though Azgp1 may improve leptin signaling acting as an antiinflammatory agent, since it may inhibit amine oxidase copper-containing 3 (Open Biol. 10: 190035.

<http://dx.doi.org/10.1098/rsob.190035>), the physiological expression of Azgp1 in the hypothalamus does not seem to be relevant. In fact the authors have employed stereotaxic surgery to allow the local expression of the protein.

In conclusion, even though the work could have been carried out correctly, unfortunately, there is not enough data in the literature to warrant this approach.

Reviewer #2 (Remarks to the Author):

Comments:

1. Line 31: reduced food intake and raised energy expenditure.
2. Line 38: I think you mean block ubiquitination and degradation.
3. Last paragraph needs a little more detail about the main findings and conclusion of the study.
4. Line 95: How was the blood assay validated. Levels are very high (mg/L range).
5. Fig 1C: Not clear what the 2 lower images are supposed to show. The bar graph panel is not cited in the legend.
6. In the refeeding experiments, one might expect the opposite results, namely fasting raising

Azgp1 and refeeding lowering based upon the findings with HFD and regular diet.

7. Many of the supplemental figures are primary, not supporting data and should be in the main manuscript (eg. Supp Fig 1 and 2).

8. Line 123-124: Should mention on normal calorie diet.

9. Line 125: Fig 1 panels cited do not match the text, which just talks about NCD.

10. Panel 1H: Why are animals on HFD eating less. Should express as energy intake rather than food weight.

11. Line 127: Body temperature results not mentioned.

12. For images showing UCP1 staining in BAT throughout the paper the differences in expression cited are not convincing. Would do RNA levels.

13. Line 136: PKA and HSL phosphorylation was affected, not total protein levels in 1Q.

14. While food intake is shown as total intake, energy expenditure is normalized to body weight, which may be misleading due to differences in BW between groups. Should show as linear regression plots of EE vs. BW. Temperature data however does tend to support what the authors state concerning EE.

15. Line 177: I do not see any WAT volume data shown.

16. Line 187: should read: effect of Azgp1 signaling in hypothalamus on glucose metabolism.

17. Line 239: Fig 6E does not show any NCD data.

18. Line 244: Hard to conclude leptin signaling per se affected as the baselines were different and increases after leptin were similar. Similar in Fig 6A, where leptin response was not affected, just baseline.

19. Line 282: I guess leptin was not administered to MBH but rather added to a brain slice including MBH.

20. Line 328: Why does AGK band pulled down by anti-Azgp1 run differently in panel 7E compared to input and IgG?

21. In Fig 8C the differences in degradation between the 2 conditions is not obvious. Should be quantified.

22. Line 363: In many assays leptin still shows responses and there are no direct data linking changes in AGK to effects on leptin signaling. Perhaps one could KO AGK and see if this blocks Azgp1 effects on leptin.

23. Line 420: Effects on insulin signaling were not addressed by experiments.

24. Line 427: Speculative, these changes could just be secondary to changes in adiposity.

Reviewer #3 (Remarks to the Author):

In this study, the authors have conducted extensive research to investigate the role of ZAG1 in POMC neurons using various biological and genetic approaches. The physiological consequences of ZAG1 over-expression or knockout were thoroughly examined, including food intake, energy expenditure, adipocyte size, glucose and insulin sensitivity. The authors conclude that *Azgp1* regulates glucose/lipid metabolism through its action on hypothalamic POMC neurons. However, the effects of *Azgp1* on glucose/lipid metabolism could be results of being obese or lean in these animals rather than the direct biological consequences of modulating ZAG1 expression in POMC neurons.

Interestingly, a recent publication titled "The melanocortin action is biased toward protection from weight loss in mice" suggested that over-expression of MC4R, POMC, or its derived peptides, which enhance melanocortin action, had little effect on preventing or reversing obesity. In contrast, the current study proposes that over-expression of ZAG1, which promotes POMC expression, ameliorates metabolic disorders in obese animals. This raises the question of the discrepancies between these findings.

To improve the manuscript, I recommend that the authors re-write and re-organize the data, addressing the following issues:

1) The novelty of the study is not clear, and there is redundant information in the main figure. For example, Figure 1 presents data on reduced circulating ZAG1 in human obesity and obese mice, which is already well-established. Consider moving this information to the supplementary figure if necessary.

2) Based on reference [17], ZAG1 expression appears to be age-dependent and cell type-

dependent. What is the overall expression pattern of ZAG1 during development, particularly in POMC neurons?

3) In Figure 2, what percentage of POMC cells express ZAG1? Additionally, what percentage of AgRP cells express ZAG1?

4) In Supplementary Figure 8, it would be valuable to show the colocalization of ZAG1 with POMC and ZAG1 with cfos.

5) I suggest the authors re-write the discussion, specifically from line 422 to line 430, as the current study does not support the descriptions made. Again, the changes in glucose and lipid metabolism could be secondary effects of obesity.

6) The authors propose that the Azgp1-AGK interaction is necessary for the regulatory action of Azgp1. Is this regulatory mechanism specific to POMC neurons?

7) Rectal temperature was consistently measured in all experiments. What is the significance of the temperature changes in these animals? Does over-expression of Azgp1 in POMC neurons modulate sympathetic output to thermogenic fat?

Other major issues.

Most of the IF figures are at very low resolution, it is very hard to ascertain the colocalization.

The animal numbers are limited in this study from 3 to 6, which is not enough for in vivo experiments.

What is the sex of the animals used in the experiments? Do the authors consider sex difference?

What's the circulating ZAG1 levels in the POMC-Azgp1-OE DIO mice.

In addition to these major issues, there are some minor concerns:

- Most of the immunofluorescence figures lack scale bars. Please add scale bars for better interpretation.

- Follow the HGNC guidelines for naming and formatting genes and proteins between humans and mice. Numerous mistakes in gene and protein names are present in the manuscript.

Response Letter (NCOMMS-23-16703A)

We are grateful for the constructive comments from the Editor and the Reviewers on our manuscript entitled “AZGP1 in POMC neurons modulates energy homeostasis and metabolism through leptin-mediated STAT3 phosphorylation”. These comments enriched the manuscript (MS) content and greatly improved its quality. We have revised the MS, changed and added figures. All changes are highlighted in red. For details, please find our point-by-point responses below.

Reviewer #1 (Remarks to the Author):

The manuscript by Qiu et al: Azgp1 in POMC neuron modulates energy homeostasis through leptin-mediated STAT3 phosphorylation is a great effort to link the glycoprotein Azgp1 to the leptin-mediated central control of metabolism and food intake.

The major flaw of the manuscript is the rationale to consider the hypothesis of the ARC nucleus as a target of the plasma protein. Even though Azgp1 has been found in CSF and brain in some neurological diseases, such as Krabbe or epilepsy, negative results have been found in control brain. Therefore, even though Azgp1 may improve leptin signaling acting as an antiinflammatory agent, since it may inhibit amine oxidase copper-containing 3 (Open Biol. 10: 190035.<http://dx.doi.org/10.1098/rsob.190035>), the physiological expression of Azgp1 in the hypothalamus does not seem to be relevant. In fact the authors have employed stereotaxic surgery to allow the local expression of the protein. In conclusion, even though the work could have been carried out correctly, unfortunately, there is not enough data in the literature to warrant this approach.

Response: We appreciate and thank this reviewer for their constructive comments. We believe that ARC is the main nuclear group acting on AZGP1 based on the following reasons and experimental results. First, the neuronal population in the ARC has been identified as a key regulatory factor for systemic metabolism. Some mechanisms related to obesity have been confirmed, and these mechanisms mainly exist in the ARC [1]. Second, we injected AAV-*Azgp1* into the hypothalamic MBH region and found that overexpression of AZGP1 significantly inhibited food intake and promoted energy metabolism in mice. Furthermore, we

found that overexpression of AZGP1 in the hypothalamus resulted in a significant increase in c-Fos expression, mainly concentrated in the ARC region, indicating a significant increase in neuronal excitability in this region. Therefore, this further suggests that AZGP1 may mainly act on the hypothalamic ARC to regulate feeding and metabolism.

We understand Reviewer 1[#]'s concern about whether the brain expresses AZGP1 under physiological conditions. Indeed, Maślińska et al. reported that AZGP1 expression was not found in the brains of young children (2 years old) [2]. However, other literature has reported the expression of AZGP1 in the brain tissue of patients with brain injury and neurons of normal SD rats. In the epileptic state, the expression of *Azgp1* mRNA and protein decreases in the brain. Maślińska et al.'s research mainly focuses on young children (7 to 25 months old), while other research ages mainly focus on young people [3-6]. Therefore, the differential expression of AZGP1 in the brain may be mainly due to age differences [7].

To further validate the expression of AZGP1 in the brain, we first discovered the expression of *Azgp1* mRNA in the hypothalamic tissue of 8-week-old C57BL/6J mice using fluorescence in situ hybridization (FISH) (**Rebuttal Fig 1a**). Subsequently, we further measured the expression of AZGP1 in the hypothalamus of mice of different ages and found that AZGP1 was lower in the hypothalamus of young and elderly mice but higher in adult mice (see **Reviewer 3#, Question 2**). Similar to other proteins [8-10], these results suggest that AZGP1 expression is related to age in the brain.

We are also highly concerned about the fact proposed by Reviewer #1 that AZGP1 inhibits amine oxidase copper-containing 3 (AOC3). For this reason, we cited this literature [11] and discussed the topic in the manuscript. In addition, through *in vitro* and *in vivo* experiments, we found that AZGP1 does not function through AOC3 in the hypothalamus (**Rebuttal Fig. 1b-d**) (see **3rd paragraph, page 20, line 455-459**), indicating the possibility of distinct mechanisms by which AZGP1 operates in different tissues.

Last, the application of stereotactic surgery in our research is in line with established practices in neuroscience, as evidenced by our work and that of other laboratories [12-15].

Rebuttal Fig 1 a *Azgp1* mRNA levels measured by situ hybridization (FISH) in the hypothalamus of eight-week-old male WT mice ($n = 3$; Scale bar, 50 μ m). **b** Western blots of AZGP1 and AOC3 protein expression in GT1-7 cells. **c** AOC3 expression in the hypothalamus of mice transfected with AAV9-sh*Aoc3*/shNC. **d** Total and phosphorylated JAK2 (p-JAK2/t-JAK2) and STAT3. Data are shown as the mean \pm SEM. * $p < 0.05$, ** $p < 0.001$.

References

- Brüning, J.C. & Fenselau, H. Integrative neurocircuits that control metabolism and food intake. *Science* 381, eab17398 (2023).
- Maślińska, D., Laure-Kamionowska, M. & Maśliński, S. Crosstalk in human brain between globoid cell leucodystrophy and zinc- α -2-glycoprotein (ZAG), a biomarker of lipid catabolism. *Folia Neuropathol* 51, 312-318 (2013).
- Liu, X. et al. New differentially expressed genes and differential DNA methylation underlying refractory epilepsy. *Oncotarget* 7, 87402-87416 (2016).
- Liu, Y. et al. neuronal zinc- α -2-glycoprotein is decreased in temporal lobe epilepsy in patients and rats 4.
- Liu, Y. et al. Overexpression of zinc- α -2-glycoprotein suppressed seizures and seizure-related neuroinflammation in pentylenetetrazol-kindled rats. *J Neuroinflammation* 15, 92 (2018).
- Tan, C. et al. Seizure-induced impairment in neuronal ketogenesis: Role of zinc- α -2-glycoprotein in mitochondria. *J Cell Mol Med* 24, 6833-6845 (2020).
- Wei, X. et al. Expression and Function of Zinc- α -2-Glycoprotein. *Neurosci Bull* 35, 540-550 (2019).
- Chow, M.L. et al. Age-dependent brain gene expression and copy number anomalies in autism suggest distinct pathological processes at young versus mature ages. *PLoS Genet* 8, e1002592 (2012).
- Matsuwaki, T., Asakura, R., Suzuki, M., Yamanouchi, K. & Nishihara, M. Age-dependent changes in progranulin expression in the mouse brain. *J Reprod Dev* 57, 113-119

(2011).

10. Prüss, H., Grosse, G., Brunk, I., Veh, R.W. & Ahnert-Hilger, G. Age-dependent axonal expression of potassium channel proteins during development in mouse hippocampus. *Histochem Cell Biol* **133**, 301-312 (2010).
11. Romauch, M. Zinc- α 2-glycoprotein as an inhibitor of amine oxidase copper-containing 3. *Open Biol* 2020; **10**: 190035.
12. Zhou M, Xu X, Wang H, Yang G, Yang M, Zhao X, Guo H, Song J, Zheng H, Zhu Z, Li L. Effect of central JAZF1 on glucose production is regulated by the PI3K-Akt-AMPK pathway. *FASEB J.* 2020;34(5):7058-7074.
13. Liu, H. et al. Hypothalamic Grb10 enhances leptin signalling and promotes weight loss. *Nat Metab* 5, 147-164 (2023).
14. Yuan, F. et al. Overexpression of Smad7 in hypothalamic POMC neurons disrupts glucose balance by attenuating central insulin signaling. *Mol Metab* 42, 101084 (2020).
15. Sun H, Lin W, Tang Y, Tu H, Chen T, Zhou J, Wang D, Xu Q, Niu J, Dong W, Liu S, Ni X, Yang W, Zhao Y, Ying L, Zhang J, Li X, Mohammadi M, Shen WL, Huang Z. Sustained remission of type 2 diabetes in rodents by centrally administered fibroblast growth factor 4. *Cell Metab.* 2023 Jun 6;35(6):1022-1037.

Reviewer #2 (Remarks to the Author):

Comments:

1. Line 31: reduced food intake and raised energy expenditure.

Response: As suggested, this content has been modified (**see page 2, lines 30-31**).

2. Line 38: I think you mean block ubiquitination and degradation.

Response: We appreciate the reviewer's more accurate description. This sentence has been revised (**see page 2, line 37**).

3. Last paragraph needs a little more detail about the main findings and conclusion of the study.

Response: As requested, the last paragraph of the Introduction has been described in detail (**see page 4, line 82-87**).

4. Line 95: How was the blood assay validated. Levels are very high (mg/L range).

Response: Serum AZGP1 concentration was measured using ELISA (Ray Biotech, GEORGIA, USA), and the levels of circulating AZGP1 concentration are consistent with other reports [1-5]. The blood assay was validated by spiking recovery testing. This test was performed by spiking two levels of human AZGP1 (2 ng/ml and 0.5 ng/ml) into human serum (serum/plasma was prediluted 500,000 fold), plasma (EDTA, citrate and heparin) and cell culture media (DMEM and RPMI 1640, prediluted 5 fold). The unspiked and spiked samples were tested with a standard curve at the same time. The incorporation of known concentrations of AZGP1 into serum or sample buffer for determining recovery rate (**Rebuttal Table 1**). The inter-assay and intra-assay coefficients of variation are < 12% and < 8% for AZGP1, respectively, which has been reported in the MS (**see 1st paragraphs**,

page 22, line 486-487).

Rebuttal Table 1 Recovery rate for AZGP1 concentration

Sample Type	Average % Recovery	Range (%)
Serum	78.73	73-84
Plasma	79.34	76-85
Cell culture media	74.51	70-78

References

1. Alenad, A.M. et al. Associations of zinc- α -2-glycoprotein with metabolic syndrome and its components among adult Arabs. *Sci Rep* **12**, 4908 (2022).
2. Anna, J. et al. ZAG (Zinc-Alpha 2 Glycoprotein) Serum Levels in Girls with Anorexia Nervosa. *J Clin Med* **12**(2023).
3. Xu, D. et al. Changes of Serum Zinc- α 2-Glycoprotein Level and Analysis of Its Related Factors in Gestational Diabetes Mellitus. *J Diabetes Res* **2021**, 8879786 (2021).
4. Kraemer, T.D. et al. Changes in AZGP1 Serum Levels and Correlation With Pulse Wave Velocity After Kidney Transplantation. *Front Cardiovasc Med* **8**, 692213 (2021).
5. Pelletier, C.C. et al. White adipose tissue overproduces the lipid-mobilizing factor zinc α 2-glycoprotein in chronic kidney disease. *Kidney Int* **83**, 878-886 (2013).

5. *Fig 1C: Not clear what the 2 lower images are supposed to show. The bar graph panel is not cited in the legend.*

Response: We apologize for this confusion. The 2 lower images are enlarged versions of the above image and are marked with dashed lines. In addition, Fig. 1c has been changed to Fig. 1b, and the scale size has been indicated in the figure legends (**see Fig. 1 b**).

6. *In the refeeding experiments, one might expect the opposite results, namely, fasting raising Azgp1 and refeeding lowering based upon the findings with HFD and regular diet.*

Response: We apologize for this confusing expression. The fasting and refeeding experiment reflected the effect of changes in acute nutritional status on Azgp1 expression, while HFD feeding for 12 weeks observed the changes in Azgp1 expression under obesity and insulin resistance conditions.

In fact, the opposite changes in *Azgp1* under fasted-to-fed and HFD-fed conditions are similar to the changes in POMC expression [1-6]. Therefore, we speculate that the upregulation of energy balance-related molecules, such as AZGP1, mediates *Pomc* mRNA expression, peptide release and POMC neuronal activation to inhibit food intake during refeeding.

The downregulation of AZGP1 expression by long-term HFD feeding may be due to the secondary effects of chronic inflammation and endoplasmic reticulum stress induced by HFD in the hypothalamus (7-8). AZGP1 downregulation further leads to the downregulation of POMC expression and reduced α -MSH release, thereby reducing energy expenditure and inducing obesity and insulin resistance.

References

1. Mizuno, T.M. et al. Hypothalamic pro-opiomelanocortin mRNA is reduced by fasting and [corrected] in ob/ob and db/db mice, but is stimulated by leptin. *Diabetes* 47, 294-297 (1998).
2. Quarta, C. et al. POMC neuronal heterogeneity in energy balance and beyond: an integrated view. *Nat Metab* 3, 299-308 (2021).
3. Wu, Q. et al. The temporal pattern of cfos activation in hypothalamic, cortical, and brainstem nuclei in response to fasting and refeeding in male mice. *Endocrinology* 155, 840-853 (2014).
4. Brandt, C. et al. Food Perception Primes Hepatic ER Homeostasis via Melanocortin-Dependent Control of mTOR Activation. *Cell* 175, 1321-1335.e1320 (2018).
5. Quarta, C. et al. Functional identity of hypothalamic melanocortin neurons depends on Tbx3. *Nat Metab* 1, 222-235 (2019).
6. Cakir, I. et al. Obesity induces hypothalamic endoplasmic reticulum stress and impairs proopiomelanocortin (POMC) post-translational processing. *J Biol Chem* 288, 17675-17688 (2013).
7. Cakir, I. et al. Obesity induces hypothalamic endoplasmic reticulum stress and impairs proopiomelanocortin (POMC) post-translational processing. *J Biol Chem* 288, 17675-17688 (2013).
8. De Souza, C.T. et al. Consumption of a fat-rich diet activates a proinflammatory response and induces insulin resistance in the hypothalamus. *Endocrinology* 146, 4192-4199 (2005).

7. Many of the supplemental figures are primary, not supporting data and should be in the main manuscript (eg. Sup Fig 1 and 2).

Response: We thank the reviewer #2 for this suggestion. As requested, Supplementary Fig 1 has been changed to **Fig. 2**. However, due to restriction on the number of Figure,

Supplementary Fig. 2 is still used as a supplementary figure (**see Supplementary Fig. 2**).

8. *Line 123-124: Should mention on normal calorie diet.*

Response: We thank the reviewer for this suggestion. As requested, this sentence has been revised (**see 3rd paragraph, page 6, line 131-133**).

9. *Line 125: Fig 1 panels cited do not match the text, which just talks about NCD.*

Response: We apologize for this mistake, and it has been corrected. The original Supplementary Fig. 1d-k has been changed to Fig. 2 d-e and Fig. 2 h-j (**see 3rd paragraph, page 6; Fig. 2 legends**).

10. *Panel 1H: Why are animals on HFD eating less. Should express as energy intake rather than food weight.*

Response: We understand the concern of Reviewer 2 #. We and other laboratories have found that animals are prone to develop anorexia when fed an HFD, manifested by reduced food intake [1-2]. In addition, as needed, food intake has been converted into energy intake (**See Fig. 2h, Fig. 4f, Fig. 5d and Supplementary Fig. 6e, Supplementary Fig. 8f, Supplementary Fig. 11h; 3rd paragraph, page 6, line 131; 3rd paragraph, page 8, line 183; 1st paragraphs, page 11, line 237; 2nd paragraphs, page 10, line 221; 1st paragraphs, page 14, line 316**).

References

1. Lai, Y. et al. DOCK5 regulates energy balance and hepatic insulin sensitivity by targeting mTORC1 signaling. *EMBO Rep* **21**, e49473 (2020).
2. Chen, P. et al. Administration Time and Dietary Patterns Modified the Effect of Inulin on CUMS-Induced Anxiety and Depression. *Mol Nutr Food Res* **67**, e2200566 (2023).

11. *Line 127: Body temperature results not mentioned.*

Response: We apologize for this oversight. As requested, body temperature has been added (**see 3rd paragraph, page 6, line 130**).

12. For images showing UCPI staining in BAT throughout the paper the differences in expression cited are not convincing. Would do RNA levels.

Response: We thank reviewer #2 for this comment and have added *Ucp1* mRNA expression (see Fig. 2n, Fig. 4m, Fig. 5n and Supplementary Fig. 8k; 1st paragraphs, page 9, line 186 and 1st paragraphs, page 11, line 243).

13. Line 136: PKA and HSL phosphorylation was affected, not total protein levels in IQ.

Response: We apologize for this carelessness, and it has been revised (see 1st paragraphs, page 7, line 141).

14. While food intake is shown as total intake, energy expenditure is normalized to body weight, which may be misleading due to differences in BW between groups. Should show as linear regression plots of EE vs. BW. Temperature data however does tend to support what the authors state concerning EE.

Response: We appreciate and agree with this suggestion. The linear regression plots of EE and BW have been added to the revised MS (see Fig. 2K, Fig. 4h, Fig. 5f, and Supplementary Fig. 8h).

15. Line 177: I do not see any WAT volume data shown.

Response: We apologize for this mistake, and it has been revised (see Fig. 4o-p).

16. Line 187: should read: effect of *Azgp1* signaling in hypothalamus on glucose metabolism.

Response: As requested, this sentence has been rewritten (see 2nd paragraphs, page 9, line 202).

17. Line 239: Fig 6E does not show any NCD data.

Response: We thank Reviewer #2 for pointing out this issue, and it has been revised (see Supplementary Fig. 8 and 1st paragraphs, page 12, line 260-261).

18. Line 244: Hard to conclude leptin signaling perse affected as the baselines were different

and increases after leptin were similar. Similar in Fig 6A, where leptin response was not affected, just baseline.

Response: We apologize for not clarifying this concept. Similar to previous reports [1-2], there was a difference in leptin signaling (POMC-STAT3) between *Azgp1^{fl/fl}* and *Azgp1* KO mice at baseline, indicating that *Azgp1* deficiency inhibits the effect of endogenous leptin on STAT3 in POMC neurons. When stimulated by exogenous leptin, this effect is further amplified. (If the reviewer considers that we did not accurately understand this comment, please let us know for further modifications).

References

1. Dodd, G.T. et al. Intranasal Targeting of Hypothalamic PTP1B and TCPTP Reinstates Leptin and Insulin Sensitivity and Promotes Weight Loss in Obesity. *Cell Rep* **28**, 2905-2922.e2905 (2019).
2. Yan, H. et al. Estrogen Improves Insulin Sensitivity and Suppresses Gluconeogenesis via the Transcription Factor Foxo1. *Diabetes* **68**, 291-304 (2019).

19. Line 282: I guess leptin was not administered to MBH but rather added to a brain slice including MBH.

Response: We appreciate this professional comment from the reviewer, and the content has been further described (see 2nd paragraphs, page 13, line 302-304).

20. Line 328: Why does AGK band pulled down by anti-Azgp1 run differently in panel 7E compared to input and IgG?

Response: We appreciate the reviewer's professional comments. This problem has been solved by optimizing experimental conditions, such as replacing buffer (see Fig. 8E).

21. In Fig 8C the differences in degradation between the 2 conditions is not obvious. Should be quantified.

Response: As needed, the differences in degradation between the 2 conditions were quantified (see Fig. 9C).

22. Line 363: *In many assays leptin still shows responses and there are no direct data linking changes in AGK to effects on leptin signaling. Perhaps one could KO AGK and see if this blocks Azgp1 effects on leptin.*

Response: We appreciate and agree with the reviewer's comment and have added an *in vivo* experiment. Eight-week-old male WT mice received MBH injection of AAV9-*Azgp1*/GFP or AAV9-*Azgp1*+AAV9-*shAgk* and were fed a HFD for 4 weeks followed by leptin treatment. The relationship between AZGP1 and AGK with leptin signals was further analysed (see Supplementary Fig. 12c-e, 1st paragraphs, page 15, line 332-328 and 2nd paragraphs, page 23, line 510, 512).

23. Line 420: *Effects on insulin signaling were not addressed by experiments.*

Response: We thank the reviewer for pointing this out. Because this MS mainly focuses on the impact of AZGP1 on leptin and its signaling system, we have removed this sentence (see 3rd paragraphs, page 19).

24. Line 427: *Speculative, these changes could just be secondary to changes in adiposity.*

Response: Determining whether the effects of hypothalamic AZGP1 on glucose/lipid metabolism are due to differences in body weight is an important question. Therefore, we set up new cohort of POMC-*Azgp1*-OE and control mice, which were fed a high-fat diet (HFD) for 4 weeks. We then performed TG, GTT and ITT tests to investigate the effects of AZGP1 on glucose/lipid metabolism before body weight divergence and found that AZGP1 overexpression did not improve TG, glucose tolerance or insulin sensitivity, indicating that the regulation of glucose/lipid metabolism by AZGP1 may be secondary to changes in body weight. (See Supplementary Fig. 4h-l, 1st paragraphs, page 10, line 210-215, and 3rd paragraph, page 22, line 503). In addition, this conclusion has been rewritten in the discussion (see 3rd paragraph, page 19, line 444-447).

Reviewer #3 (Remarks to the Author):

*In this study, the authors conducted extensive research to investigate the role of ZAG1 in POMC neurons using various biological and genetic approaches. The physiological consequences of ZAG1 over-expression or knockout were thoroughly examined, including food intake, energy expenditure, adipocyte size, glucose and insulin sensitivity. The authors conclude that *Azgp1* regulates glucose/lipid metabolism through its action on hypothalamic POMC neurons.*

*However, the effects of *Azgp1* on glucose/lipid metabolism could be results of being obese or lean in these animals rather than the direct biological consequences of modulating ZAG1 expression in POMC neurons.*

Response: We appreciate the positive evaluation from Reviewer 3 # on this MS and agree with the reviewer's view that the impact of AZGP1 on glucose/lipid metabolism may be due to changes in body weight. We added an experiment in POMC-*Azgp1*-OE and control mice fed a HFD for 4 weeks. We then performed TG, GTT and ITT tests to investigate the effects of AZGP1 on glucose/lipid metabolism before body weight divergence and found that AZGP1 overexpression did not improve TG, glucose tolerance or insulin sensitivity, indicating that the regulation of glucose/lipid metabolism by AZGP1 may be secondary to changes in body weight. (See Supplementary Fig. 4h-l, 1st paragraphs, page 10, line 210-215, and 3rd paragraph, page 22, line 503). In addition, this conclusion has been rewritten in the discussion (see 3rd paragraph, page 19, line 444-447).

Interestingly, a recent publication titled "The melanocortin action is biased toward protection from weight loss in mice" suggested that over-expression of MC4R, POMC, or its derived peptides, which enhance melanocortin action, had little effect on preventing or reversing obesity. In contrast, the current study proposes that over-expression of ZAG1, which promotes POMC expression, ameliorates metabolic disorders in obese animals. This raises the question of the discrepancies between these findings.

Response: We appreciate the constructive comments from reviewer #3, and this issue has been added to the discussion (see 1st paragraphs, page 18, line 409-414).

To improve the manuscript, I recommend that the authors re-write and re-organize the data, addressing the following issues:

1) The novelty of the study is not clear, and there is redundant information in the main figure. For example, Figure 1 presents data on reduced circulating ZAG1 in human obesity and obese mice, which is already well-established. Consider moving this information to the supplementary figure if necessary.

2) Response: We appreciate and agree with the reviewer's suggestions. To highlight the novelty of the study, as requested, the data on reduced circulating AZGP1 in human obesity in Figure 1 have been moved to Supplementary Fig. 1. However, data on the expression of *Azgp1* in the hypothalamus of obese mice have not been reported. Therefore, we are considering keeping them in the main Figure (**see Supplementary Fig. 1 and 1st paragraphs, page 5, line 96-97**).

2) Based on reference [17], ZAG1 expression appears to be age-dependent and cell type-dependent. What is the overall expression pattern of ZAG1 during development, particularly in POMC neurons?

Response: We thank the reviewer for this insightful comment. To address this issue, we examined AZGP1 protein expression in the hypothalamus of C57BL/6J mice of different ages and found that AZGP1 expression was lower in the hypothalami of young mice (1, 5, 10, 20 days) and older mice (18 and 24 months), while higher expression was observed in mice aged 1-12 months. Therefore, these results suggest that AZGP1 expression in the hypothalamus is age-related (**see Fig. 1a and 1st paragraphs, page 5, line 98-103**). In addition, AZGP1 expression in POMC neurons was consistent with its expression in the hypothalamus (**see Supplementary Fig. 3a and 2nd paragraphs, page 7, line 163-165**).

3) In Figure 2, what percentage of POMC cells express ZAG1? Additionally, what percentage of AgRP cells express ZAG1?

Response: We thank Reviewer 3# for this constructive comment. As requested, this content

has been added to the MS (see Fig. 3a and ^{2nd} paragraph, page 7, line 159-163).

4) *In Supplementary Figure 8, it would be valuable to show the colocalization of ZAG1 with POMC and ZAG1 with cfos.*

Response: As suggested, the colocalization of AZGP1 with POMC and AZGP1 with c-Fos are shown in Supplementary Fig. 9 e and f, and the results are described in the revised MS (see 1st paragraphs, page 12, line 261-264).

5) *I suggest the authors re-write the discussion, specifically from line 422 to line 430, as the current study does not support the descriptions made. Again, the changes in glucose and lipid metabolism could be secondary effects of obesity.*

Response: We appreciate the suggestion from Reviewer 3 #. These contents (line 422 to line 430) have been deleted, and the discussion section has been modified (see the discussion).

6) *The authors propose that the Azgp1-AGK interaction is necessary for the regulatory action of Azgp1. Is this regulatory mechanism specific to POMC neurons?*

Response: We speculate that this regulatory mechanism may not be specific to POMC neurons. However, we believe that the Azgp1-AGK interaction may be specific to the regulation of metabolism by Azgp1, at least in the hypothalamus.

7) *Rectal temperature was consistently measured in all experiments. What is the significance of the temperature changes in these animals? Does over-expression of Azgp1 in POMC neurons modulate sympathetic output to thermogenic fat?*

Response: Rectal temperature can reflect thermogenesis in mice and serve as an indirect indicator for evaluating energy consumption [1-2].

To evaluate whether overexpression of Azgp1 in POMC neurons modulates sympathetic output to thermogenic fat, we examined the norepinephrine (NE) receptor (Adrb3) and tyrosine hydroxylase (TH) in BAT of mice with Azgp1 overexpression in POMC neurons and found that *Th* and *Adrb3* mRNA expression was significantly higher in BAT of mice with

AZGP1 overexpression than in control mice (**Fig. 4n**), indicating that overexpression of AZGP1 stimulated sympathetic nervous activity in BAT, leading to an increase in heat production in BAT [3] (**see Fig. 4n and 1st paragraphs, page 9, line 188-193**).

References

1. Jiang, L. et al. Leptin receptor-expressing neuron Sh2b1 supports sympathetic nervous system and protects against obesity and metabolic disease. *Nat Commun* **11**, 1517 (2020).
2. 28. Wang, P. et al. A leptin-BDNF pathway regulating sympathetic innervation of adipose tissue. *Nature* **583**, 839-844 (2020).
3. Tang, Q. et al. MANF in POMC Neurons Promotes Brown Adipose Tissue Thermogenesis and Protects Against Diet-Induced Obesity. *Diabetes* **71**, 2344-2359 (2022).

Other major issues.

1. *Most of the IF figures are at very low resolution, it is very hard to ascertain the colocalization.*

Response: Thank you for pointing out this issue. To ascertain the colocalization information more clearly, we used confocal imaging at higher magnification (**see Fig. 3a and d, Fig. 6f, Fig. 8b, Fig. 8d, Supplementary Fig. 7a, Supplementary Fig. 12b, g and h**).

S

2. *The animal numbers are limited in this study from 3 to 6, which is not enough for in vivo experiments.*

Response: We apologize for not describing this clearly. Most of our experiments used 7-10 mice in each group. For the experimental group with fewer than 5 mice, we increased the number of mice (**Fig. 5**). In addition, we also indicated the number of mice in the annotation.

3. *What is the sex of the animals used in the experiments? Do the authors consider sex difference?*

Response: We apologize for not providing a detailed description. We annotated gender and added data for female mice in the revised MS (**Supplementary Fig. 5 and 1st paragraphs, page 10, line 214-215**).

3. What's the circulating ZAG1 levels in the POMC-Azgp1-OE DIO mice?

Response: As suggested, we measured the serum Azgp1 concentration using ELISA (SMK6273B, Mei Ke, Jiangsu, China) and found no difference between POMC-Azgp1-OE mice and the control group, indicating that overexpression of Azgp1 in POMC neurons has little effect on its circulating concentration (**Rebuttal Fig. 2**).

Rebuttal Fig. 2 Serum AZGP1 concentration in male POMC-Azgp1-OE DIO and control mice (n = 9). Data are shown as mean \pm SEM.

In addition to these major issues, there are some minor concerns:

1) - *Most of the immunofluorescence figures lack scale bars. Please add scale bars for better interpretation.*

Response: We are very grateful for the reviewer's reminder that the scale bars have been added to all immunofluorescence figures.

2) - *Follow the HGNC guidelines for naming and formatting genes and proteins between humans and mice. Numerous mistakes in gene and protein names are present in the manuscript.*

Response: We apologize for these errors and thank Reviewer #3 for pointing them out. According to the HGNC guidelines, we have revised the naming mistakes of these genes and proteins.

REVIEWER COMMENTS

Reviewer #2 (Remarks to the Author):

My concerns have been addressed. I have only one minor comment. In Supp Fig 1 the figure legend says serum levels were measured but the x axis is labeled plasma.

Reviewer #3 (Remarks to the Author):

While I acknowledge the authors' diligent efforts to address my concerns in their revisions, some queries regarding the manuscript persist despite these careful adjustments.

1. Line 100: "AZGP1 expression was lower in very young mice (1-20 days)" should be expressed as "postnatal day 1 (P1) to 20 (P20)". Similarly, Figure 1a should be corrected to reflect P1, P5...
2. Line 103: I would suggest adding a similar sentence like "The current study was performed in mice aged 8-12 weeks, a period coinciding with abundant AZGP1 expression in the hypothalamus." to connect the developmental study to your experiment.
3. Lines 104 – 111: Did the authors investigate AZGP1 expression in other nuclei associated with metabolism, such as PVH and NTS? In other words, was AZGP1 expression thoroughly examined across the entire brain under the NCD and HFD condition? Are any other brain regions exhibit differential expression of AZGP1?
4. To focus on the ARC and VMH, the authors might consider specifying these areas rather than the entire hypothalamus in Figures 1C to 1G.
5. Lines 116-118: Except for Figure 2b, all experiments utilized entire hypothalamic tissue. Thus, it is challenging to conclusively state, "Taken together, these results indicate that AZGP1 is abundantly expressed in the hypothalamic ARC."
6. Figure 3d, I am very shocked by the POMC staining in the azgp1+leptin treatment. It

seems like azgp1 and leptin have synergistic effects to increase the number of POMC cells, it also seems like almost all the cells in this panel expressing POMC if you compare DAPI and POMC staining? Another concern is that if you compare DAPI staining in the Azgp1+aCSF treatment with other three treatment, less number of cells in this panel, however, they are using the same scale bar. Any explanation to all these questions?

7. Lines 176-177: How does Azgp1 overexpression in the hypothalamic ARC increase POMC expression? Does it involve an increase in the number of POMC+ cells?

8. Figure 7a and 7b: The 3V labeling seems to overlap with the Pomc staining. Consider relocating this labeling for clarity.

Please find our responses point-by-point responses below.

Reviewer #2 (Remarks to the Author):

My concerns have been addressed. I have only one minor comment. In Supp Fig 1 the figure legend says serum levels were measured but the x axis is labeled plasma.

Response: We appreciate your approval of this revised MS. Furthermore, we apologize for this error and it has been corrected (see **Supp Fig 1**).

Reviewer #3 (Remarks to the Author):

While I acknowledge the authors' diligent efforts to address my concerns in their revisions, some queries regarding the manuscript persist despite these careful adjustments.

1. Line 100: "AZGP1 expression was lower in very young mice (1-20 days)" should be expressed as "postnatal day 1 (P1) to 20 (P20)". Similarly, Figure 1a should be corrected to reflect P1, P5...

Response: Thank you for your affirmation of our previous revision work. Furthermore, based on your suggestion, we have revised this statement (see line 100 and Fig. 1a).

2. Line 103: I would suggest adding a similar sentence like "The current study was performed in mice aged 8-12 weeks, a period coinciding with abundant AZGP1 expression in the hypothalamus." to connect the developmental study to your experiment.

Response: Thank you for this valuable suggestion, and this content has been added to the revised MS (see line 103-104).

3. Lines 104 – 111: *Did the authors investigate AZGP1 expression in other nuclei associated with metabolism, such as PVH and NTS? In other words, was AZGP1 expression thoroughly examined across the entire brain under the NCD and HFD condition? Are any other brain regions exhibit differential expression of AZGP1?*

Response: Based on your suggestion, we examined the expression of *Azgp1* in various brain regions. We discovered that *Azgp1* is primarily expressed in the cortex, hippocampus, cerebellum, and hypothalamus. Furthermore, we observed no significant differences in the expression of *Azgp1* in the PVH, NTS, or other brain regions when comparing NCD and HFD feeding conditions (see **Rebuttal Fig 1 a-f**).

Rebuttal Fig 1 AZGP1 expression in the brains of NCD- and HFD-fed mice. **(A)** IF staining

for AZGP1 expression in PVH. **(B)** IF staining for AZGP1 expression in cerebellum. **(C)** IF staining for AZGP1 expression in cerebral cortex. **(D)** IF staining for AZGP1 expression in NTS. **(E)** IF staining for AZGP1 expression in hippocampal. **(F)** AZGP1 protein expression and quantification in the brain. NCD, normal chow diet; HFD, high-fat diet; PVH, paraventricular; ML, molecular layer, PCL, Purkinje cell. gl, internal granule layer. NTS, nucleus tractus solitarius; CA1, cornu ammonis1; AP, the area postrema; DG, dentate gyrus. NS, not significant. Data are expressed as the mean \pm SEM.

4. To focus on the ARC and VMH, the authors might consider specifying these areas rather than the entire hypothalamus in Figures 1C to 1G.

Response: Thank you for your reminder. We added western blot analysis of AZGP1 expression in the MBH (ARC+VMH) and found that the results were consistent with the analysis of the entire hypothalamus (see Fig. 1C-G and page 5-6 and page 25).

5. Lines 116-118: Except for Figure 2b, all experiments utilized entire hypothalamic tissue. Thus, it is challenging to conclusively state, "Taken together, these results indicate that AZGP1 is abundantly expressed in the hypothalamic ARC."

Response: We appreciate your reminder. This description has been corrected (see page 6).

6. Figure 3d, I am very shocked by the POMC staining in the azgp1+leptin treatment. It seems like azgp1 and leptin have synergistic effects to increase the number of POMC cells, it also seems like almost all the cells in this panel expressing POMC if you compare DAPI and POMC staining? Another concern is that if you compare DAPI staining in the Azgp1+aCSF treatment with other three treatment, less number of cells in this panel, however, they are using the same scale bar. Any explanation to all these questions?

Response: We understand your concerns. We found that AZGP1 overexpression increased POMC protein expression (not entirely equivalent to an increase in the number of POMC cells), which was more pronounced under leptin stimulation. The IF staining shown is from a representative section selected for display. In other sections, it can be seen that not all cells express POMC protein (**Rebuttal Fig. 2**). We apologize for the issue of less number of cells in

the *Azgp1*+ aCSF panel, which was caused by the selection of slices at different magnifications. We have selected images at the same magnification in the revised MS (see **Figure 3d**). In addition, we also examined the staining results of other mice in this group and did not observe a decrease in cell number (**Rebuttal Fig. 3**).

Rebuttle Fig. 2 IF staining of POMC and c-Fos in the mouse ARC (n = 3; scale bar: 50 μm). 3V, third cerebral ventricle; ARC, arcuate nucleus.

Rebuttle Fig. 3 IF staining of POMC and c-Fos in the mouse ARC (n = 3; scale bar: 50 μm). 3V, third cerebral ventricle; ARC, arcuate nucleus.

7. Lines 176-177: How does *Azgp1* overexpression in the hypothalamic ARC increase POMC expression? Does it involve an increase in the number of POMC+ cells?

Response: We apologize for this confusion. We believe that this difference may be due to *Azgp1* overexpression leading to increased excitability of POMC neurons and thereby promoting POMC expression. In addition, our results showed that AZGP1 expression did not alter the number of POMC neurons (POMC-tdTomato) (see **Rebuttal Fig4**).

Rebuttle Fig 4 POMC neurons (tdTomato) of POMC-*Azgp1* KO or POMC-Cre^{ER} mice fed a HFD for 12 weeks. (n = 3 slices; scale bars: 50 μ m). 3V, third cerebral ventricle; ARC, arcuate nucleus. tdTomato, POMC. Data were shown as mean \pm SEM.

8. *Figure 7a and 7b: The 3V labeling seems to overlap with the Pomc staining. Consider relocating this labeling for clarity.*

Response: We appreciate your suggestion, and this label has been relocated (see Fig.7a and b).

REVIEWERS' COMMENTS

Reviewer #3 (Remarks to the Author):

Thank you to the authors for their meticulous revisions; my concerns have been fully addressed. However, I would suggest the authors to carefully review the manuscript for typos, eg, line 565, there is a "=".

Please find below our point-by-point responses to the comments of Reviewer #3:

Reviewer #3 (Remarks to the Author):

Thank you to the authors for their meticulous revisions; my concerns have been fully addressed. However, I would suggest the authors to carefully review the manuscript for typos, eg, line 565, there is a "=".

Response: We appreciate the positive feedback from Reviewer #3 and are pleased to hear that the concerns raised in the previous review have been fully addressed. We thank the reviewer for pointing out the typographical error in line 565 of the manuscript. We have carefully reviewed the entire manuscript again for any additional typos and have corrected the “=” sign error in line 565. We have also implemented a thorough proofreading process to ensure that such errors are minimized in the final version of the manuscript. We are grateful for the reviewer’s attention to detail.